# Inversion Algorithm of Black Carbon Mixing State Based on Machine Learning

Zeyuan Tian[1,2], Jiandong Wang[1,2], Jiaping Wang[3,4], Chao Liu[1,2], Jia Xing[5], Jinbo Wang[3,4], Zhouyang Zhang[1,2], Yuzhi Jin[1,2], Sunan Shen[1,2], Bin Wang[1,2], Wei Nie[3,4], Xin Huang[3,4], Aijun Ding[3,4]

[1]Collaborative Innovation Center on Forecast and Evaluation of Meteorological Disasters, Nanjing University of Information Science and Technology, Nanjing, China,
[2]China Meteorological Administration Aerosol-Cloud-Precipitation Key Laboratory, School of Atmospheric Physics, Nanjing University of Information Science and Technology, Nanjing, China,
[3]Joint International Research Laboratory of Atmospheric and Earth System Sciences, School of Atmospheric Sciences, Nanjing University, Nanjing, China,
[4]National Observation and Research Station for Atmospheric Processes and Environmental Change in Yangtze River Delta, Nanjing, China,
[5]Department of Civil and Environmental Engineering, University of Tennessee, Knoxville, 37996, USA

*Correspondence to*: Jiandong Wang (jiandong.wang@nuist.edu.cn)

**Abstract.** Black carbon (BC) radiative properties are significantly influenced by its mixing state. The single-particle soot photometer (SP2) is a widely recognized instrument for quantifying BC mixing state. However, the derivation of BC mixing state from SP2 is quite challenging. Since the SP2 records individual particle signals, it requires complex data processing to convert raw signals into particle size and mixing states. Besides, the rapid accumulation of substantial data volumes impedes real-time analysis of BC mixing states. This study employs a light gradient boosting machine (LightGBM), an advanced tree-based ensemble learning algorithm, to establish an inversion model that directly correlates SP2 signals with the mixing state of BC-containing particles. Our model achieves high accuracy for both particle size inversion and optical cross-section inversion of BC-containing particles, with coefficient of determination $R^2$ higher than 0.98. We further employed the SHapley Additive exPlanation (SHAP) method to analyze the importance of input features from SP2 signals in the inversion model of the entire particle diameter ($D_p$) and explored their underlying physical significance. Compared to the widely used Leading-Edge-Only (LEO) fitting method, the machine learning (ML) method utilizes a larger coverage of signals encompassing the peak of scattering signal rather than the leading-edge data. This allows for more accurate capture of the diverse characteristics of particles. Moreover, the ML method uses signals with a high signal-to-noise ratio, providing better noise resistance. Our model is capable of accurately and efficiently acquiring the single-particle information and statistical results of the BC mixing state, which provides essential data for BC aging mechanism investigation and further BC radiative effects assessment.

## 1 Introduction

Black carbon (BC) is the dominant absorbing aerosol, making it an important contributor to positive radiative forcing in the present-day atmosphere (Bond et al., 2013; Bond and Bergstrom, 2006; Fierce et al., 2020; Liu et al., 2017; Matsui et al., 2018;

Ramanathan and Carmichael, 2008). As the product of incomplete combustion of fossil fuel combustion and biomass burning activities (Jacobson, 2001), BC is refractory with a vaporization temperature near 4000 K. During atmospheric transport, freshly emitted BC evolves from an externally mixed state to an internally mixed structure due to coagulation and condensation with other aerosol components. Changes in the mixing state can alter the light absorption and other properties of BC, thereby affecting its climate effect. For example, the presence of coating materials on BC can increase its mass absorption cross-section (MAC) relative to uncoated BC by lensing effect (Bond and Bergstrom, 2006; Cappa et al., 2012; Fuller et al., 1999). Therefore, identifying the mixing states of BC-containing particles and their relative abundance is essential for evaluating their climate effects.

The single-particle soot photometer (SP2) is a well-recognized instrument that can be used for measuring the mixing state of BC (Moteki and Kondo, 2007; Schwarz et al., 2006; Sedlacek et al., 2012; Stephens et al., 2003). By analyzing the signals observed by SP2, quantitative characterization of the mixing states of BC can be obtained. Because of the vaporization of the non-refractory material in BC-containing particle, the scattering signals of BC-containing particles obtained by SP2 will be distorted, which poses significant difficulties in defining the original particle size ($D_p$). The leading-edge-only (LEO) fitting method is widely used (Gao et al., 2007; Moteki and Kondo, 2008; Schwarz et al., 2008) to obtain $D_p$, wherein the complete Gaussian function is reconstructed by fitting the scattering signal before particle vaporization (Liu et al., 2014; Shiraiwa et al., 2008; Zhang et al., 2016). Since SP2 can track the incandescence and scattering signal of each particle, field observation using SP2 will generate a large amount of data. Performing physical inversion of particle size requires complex data processing and fitting processes, making it difficult to obtain real-time online BC mixing states.

As an alternative, data-driven models such as machine learning (ML) can provide a good supplement to physical process-based models. ML can efficiently capture the nonlinear relationship between inputs and outputs, and has found widespread application in various fields (Carleo et al., 2019; Jordan and Mitchell, 2015; Liakos et al., 2018; Tarca et al., 2007). In recent years, tree-based machine learning models have gained considerable popularity due to their extremely high computational speed, satisfactory accuracy, and interpretability (Keller and Evans, 2019; Li et al., 2022; Wei et al., 2021; Yang et al., 2020). Among these, the Light Gradient Boosting Machine (LightGBM) has shown particularly outstanding performance. As a novel distributed gradient boosting framework based on decision tree algorithms, LightGBM can extract information from data more effectively than traditional tree models, excelling in handling complex non-linear relationships and high-dimensional features (Ke et al., 2017; Liu et al., 2024; Zhong et al., 2021). It employs innovative techniques such as gradient-based one-side sampling (GOSS) and exclusive feature bundling (EFB), which significantly improve computational efficiency while maintaining high predictive performance (Ke et al., 2017; Sun et al., 2020). Furthermore, different from some black-box models, LightGBM maintains the interpretability characteristic of tree-based models (Gan et al., 2021; Zhang et al., 2019), which can provide decision path analysis, allowing for deeper insights into the decision-making process. Considering these advantages, LightGBM can be an ideal tool for analyzing large SP2 datasets and inverting BC mixing states.

In this study, an inversion model based on LightGBM is developed to establish a relationship between the SP2 time-dependent signals and both the size and optical characteristics of individual BC-containing particles. This method can simplify the process of quantitative analysis of BC mixing states, making it capable of real-time mixing state analysis. In addition, the SHapley Additive explanation (SHAP) approach is incorporated to interpret the developed model, providing insights into how different signal features contribute to the model's predictions. The LightGBM-based inversion model is then compared with the LEO fitting method to validate its performance. Finally, through this innovative modeling approach, the mixing state characteristics of BC-containing particles can be analyzed in detail, including both single-particle scale and a bulk of BC-containing particles.

## 2 Method

### 2.1 Experimental site

The SP2 observational data used in this study is from 1 April 2022 to 31 May 2022 at SORPES (Station for Observing Regional Process of the Earth system) station, which is located in Xianlin Campus of Nanjing University in Nanjing, Jiangsu Province (a regional background site in the Yangtze River Delta region in China).

### 2.2 SP2 apparatus and detection principle

The SP2 consists of an intracavity Nd:YAG laser and four optical detectors. The laser operates in a TEM00 mode, with a Gaussian intensity distribution. The laser intensity within the cavity is approximately $10^6$ W cm$^{-2}$, which is sufficient to vaporize absorbing particles as they pass through the beam (Stephens et al., 2003). The refractory particle absorbs light and has a high vaporization temperature. When heated by the laser beam to the boiling point (about 4000 K), it emits visible thermal radiation ("incandescent light"). The intensity of this thermal radiation depends on the composition and mass of the refractory components, regardless of the particle morphology and mixing state (Schwarz et al., 2006; Slowik et al., 2007). Pure scattering particles cannot absorb enough energy to heat themselves and therefore do not emit incandescent light. Particle size, therefore, can be measured based on the amount of light they scatter from the laser, which exhibits a Gaussian dependence with time (Gao et al., 2007).

Four optical detectors are synchronously sampled at 2.5 MHz. One avalanche photo-detector (APD) is optically filtered to pass only 1064 nm radiation and measures the scattering signal from all particles, including both pure scattering particles and absorbing particles. The two other APDs measure incandescence signal in the visible range, optically filtered to pass broadband light at 400–650 nm and narrowband light at 610–650 nm. The ratio of signals from these two detectors can be employed to ascertain the vaporization temperature of the particles (Schwarz et al., 2006), ensuring that the measured particle is BC. The fourth two-element APD (TEAPD) detector measures the location of leading-edge data in the laser beam, which can be used to analyze the amount of coating or mixing state of the incandescent particles.

## 3 Machine-learning-based inversion algorithm

### 3.1 Overview of the inversion model

To better apply the information obtained by SP2 as well as efficiently estimate particle size and optical properties, this study employs an innovative approach. We utilize SP2 scattering and incandescence signals as feature data, while particle size and optical properties derived through traditional physical inversion methods serve as label data. By employing the LightGBM algorithm, we establish a non-linear mapping relationship between SP2 signals and particle physical properties, aiming to provide an effective alternative to traditional inversion methods (Fig. 1).

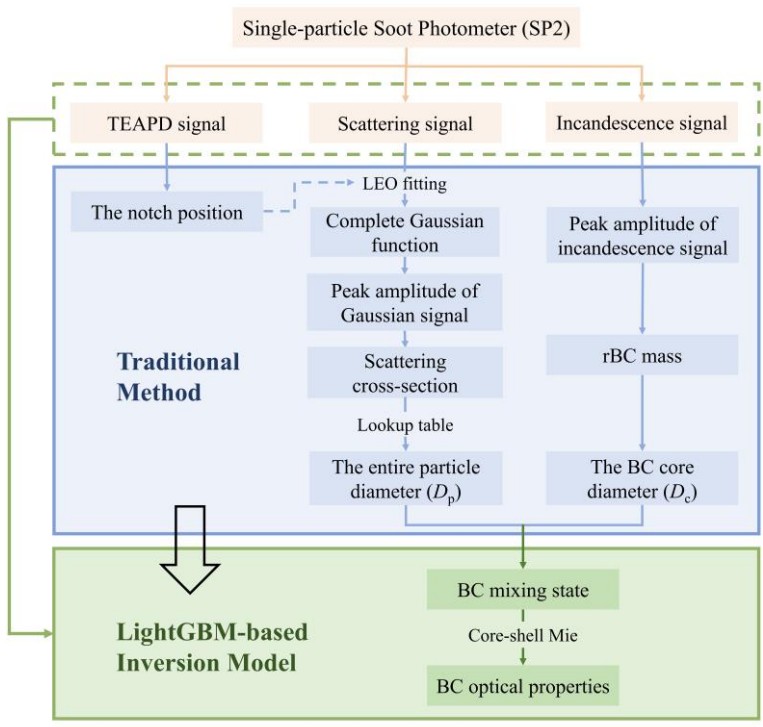

**Figure 1.** Schematic diagram of BC mixing state inversion process in this study.

### 3.2 Classification of particles measured by SP2

The physical properties of different particle types are reflected through various dimensions of SP2 signals, making particle classification essential for both SP2 data analysis and feature selection in machine learning model development. In this study, ambient particles measured by SP2 are classified into pure scattering particles and BC-containing particles. Pure scattering particles are those that only scatter light without significant absorption, while BC-containing particles, which contain refractory BC (rBC), both scatter and absorb light. BC-containing particles are further subdivided into externally mixed BC and internally mixed BC. Externally mixed BC refers to freshly emitted BC particles that have not yet mixed with other aerosol components,

while internally mixed BC describes BC that has undergone atmospheric aging processes and incorporated with other materials (Oshima et al., 2009). Operationally, we differentiate the pure scattering particle and BC-containing particle depending on whether it has the incandescence signal. When the amplitude of the incandescence signal exceeds a certain degree, the particle will be considered as BC-containing particle. Otherwise, it will be considered as pure scattering particle (Fig. 2a). The classification between externally and internally mixed BC is determined by the time delay ($\Delta t$), defined as the time difference between the peak of the incandescence signal and the scattering signal (Moteki and Kondo, 2007; Schwarz et al., 2006). The incandescence signal peak occurs when all non-BC material has evaporated and the BC reaches its incandescence temperature, thus the magnitude of $\Delta t$ correlates with the thickness of the coating on BC particles: a larger $\Delta t$ corresponds to a thicker coating that takes longer to evaporate. By examining the distribution of $\Delta t$ values in the SP2 measurements, as illustrated in Fig. 2d (Sedlacek et al., 2012; Subramanian et al., 2010; Zhang et al., 2016), BC-containing particles with $\Delta t < 2$ μs are classified as externally mixed BC (Fig. 2b), while those with $\Delta t \geq 2$ μs are categorized as internally mixed BC (Fig. 2c). It is worth noting that while some studies consider BC-containing particles with $\Delta t < 2$ μs as bare BC or thinly coated BC, we do not make this distinction in this study and treat them all as externally mixed. Additionally, relying solely on time delay may not be sufficient to distinguish certain types of BC-containing particles, such as "attached type" (Sedlacek et al., 2015). Therefore, in this study, no further classification is made regarding the detailed morphology of BC-containing particles.

Based on the aforementioned classification framework, this study develops specialized inversion models for each particle type. For pure scattering particles, the model inverts the entire particle diameter ($D_p$) and scattering cross-section ($C_{sca}$). For BC-containing particle, the model inverts the BC core diameter ($D_c$). For internally mixed BC, the model inverts $D_p$ along with both $C_{sca}$ and absorption cross-sections ($C_{abs}$). Additionally, for externally mixed BC, the model focuses on inverting $C_{sca}$ and $C_{abs}$. These targeted inversion models collectively provide a comprehensive analytical toolkit for accurate characterization of particle physical properties across different mixing states.

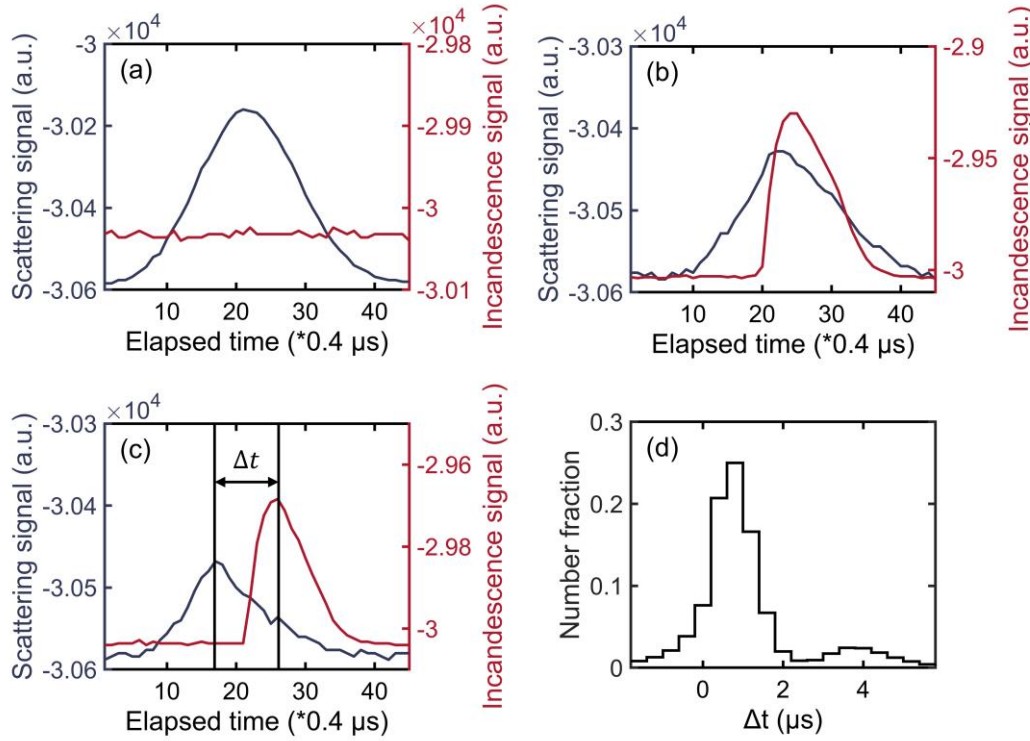

**Figure 2.** (a–c) Time series of the scattering signal and incandescence signal of different types of particles: (a) pure scattering particle; (b) externally mixed BC; (c) internally mixed BC. (d) Histogram of the time delay (Δt) of BC-containing particles. Note that the signals shown in panels (a–c) represent portions of the original signals, rather than the complete original signals. These displayed signal portions are related to the feature signal introduced in the following Sect. 3.3.

### 3.3 Construction of feature dataset

The SP2 signal is recorded based on the elapsed time, with each time window corresponding to information about a single particle. For each particle, the original scattering signal and incandescence signal are both 100-dimensional. The position of particles within the instrument is not known in advance. Additionally, the presence of instrumental background noise in the original signals would interfere the learning process and degrade ML model performance. Therefore, it is necessary to preprocess the original signals, retaining only the signal features that are most relevant to particle size and optical properties. This study implements specific feature selection methods for different particle types to extract the most informative characteristics from the original signals.

For pure scattering particles, when they traverse the laser beam, they only scatter laser light. Therefore, when constructing an ML model for pure scattering particles, scattering signals are typically used as feature data to obtain relevant information about the particles. Due to the Gaussian distribution of laser beam intensity, the scattering signal of particles passing through the laser beam is influenced by two primary factors: the particle size and laser intensity. To accurately invert particle size and

scattering cross-section, it is necessary to eliminate the influence of laser intensity distribution, which means ensuring that particles are positioned at consistent locations within the instrument for each corresponding dimension. As mentioned in Sect. 2.2, one of the four detectors in the SP2 is a split APD detector. This detector has a gap perpendicular to the particle's direction of motion, resulting in a notch in the TEAPD signal, as shown in Fig. 3a. Given the stability of SP2's optical alignment and constant sample flow rate, this notch provides a precise time reference for a particle's position within the instrument. In practice, the signal from leading element is inverted, transforming the notch into a zero-crossing point (Fig. 3b) (Gao et al., 2007). Since SP2 simultaneously records data from all four detector channels, this time reference is valid for the signals from the other three detectors as well. We locate the zero-crossing point in the scattering signal and then extract 22 data points both before and after it, creating a 45-dimensional feature dataset (Fig. S1a). Through this standardization, the differences in signal intensity can be accurately attributed to the inherent physical properties of the particles.

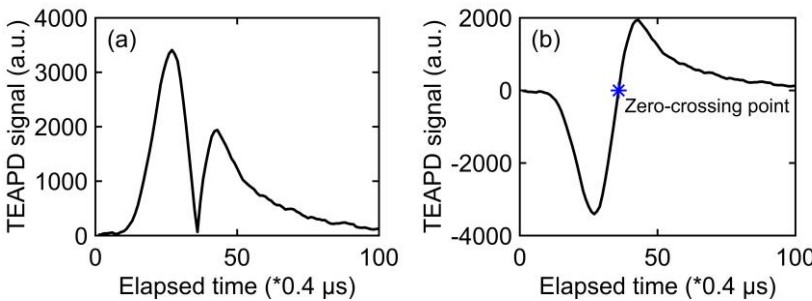

**Figure 3.** (a) The original scattering signal measured by TEAPD before the signal from the leading element is inverted. (b) The TEAPD signal obtained by SP2, with the blue asterisk indicating the position of the zero-crossing point.

When BC-containing particles pass through the laser beam, their rBC component absorbs laser energy and reaches an incandescent state, generating incandescent radiation. The peak intensity of the incandescence signal exhibits a linear correlation with the rBC mass of the particle (Moteki and Kondo, 2010). Based on this characteristic, the peak of the incandescence signal is selected as a reference point, from which 22 data points are extracted both preceding and following this point (Fig. S1b), yielding a 45-dimensional feature dataset used for inverting the $D_c$ of BC-containing particles. This method ensures the incandescence signal peaks from different BC-containing particles are positioned at the same dimension within the feature dataset, facilitating direct comparisons between particles while preserving comprehensive information about the incandescence process.

For externally mixed BC, their optical cross-section is determined by the rBC component. Consequently, when inverting the optical cross-section of externally mixed BC, we employ the same approach used for inverting $D_c$ of BC-containing particles, that is, selecting 45-dimensional feature signals centred on the peak of the incandescence signal to construct the inversion model.

Compared with other particle types, the internally mixed BC has a more complex structure. Its diameter and optical cross-section are jointly characterized by the scattering and incandescence signals produced when passing through the laser. To construct an inversion model for internally mixed BC, both types of signals are applied as feature data. For the scattering signal, we employ the same feature extraction method as used for pure scattering particles, selecting 22 data points of the scattering signal before and after the zero-crossing point. Simultaneously, considering that the relative relationship between the original

incandescence and scattering signals can reflect particle characteristics, such as coating thickness (Moteki and Kondo, 2007; Schwarz et al., 2006; Subramanian et al., 2010), the incandescence signal is selected with 22 data points before and after the zero-crossing point in a similar way (Fig. S1c). The feature extraction process yields a 90-dimensional feature dataset, comprising 45-dimensional scattering signal and 45-dimensional incandescence signal, ensuring that we can comprehensively capture the key characteristics of internally mixed BC.

**3.4 Construction of label dataset**

    In this study, the label dataset is constructed by using physical inversion methods to derive the particle size and optical properties from the corresponding SP2 signals. For pure scattering particles, when they pass through the laser, their scattering cross-section remains unchanged, resulting in an undistorted scattering signal with a Gaussian profile. Mie calculations indicate that, for spherical particles with diameters less than 1 μm, the scattering amplitude detected by SP2 exhibits a monotonic

relationship with scattering cross-section (Gao et al., 2007), and this relationship can be determined by the calibration using polystyrene latex spheres (PSL). To obtain the $D_\mathrm{p}$ of pure scattering particles, the scattering signal amplitude is first used to determine the particle's scattering cross-section, which is then compared with that of PSL particles of known diameter to determine the $D_\mathrm{p}$.

When the BC-containing particle crosses the laser, the rBC component absorbs the laser energy, causing the coating (if present) to evaporate. The absorbed energy heats the rBC until it reaches incandescence, producing detectable amounts of thermal radiation. The peak intensity of thermal radiation emitted by the rBC is proportional to its mass ($M_\mathrm{BC}$) (Moteki and Kondo, 2007). According to the empirical relationship between the incandescent light intensity and the particle mass calibrated using fullerene soot, the $M_\mathrm{BC}$ of each BC-containing particle can be quantified. Assuming a density of 1.8 g cm$^{-3}$ (Bond and

Bergstrom, 2006), the measured $M_\mathrm{BC}$ can be further converted into the mass-equivalent diameter $D_\mathrm{c}$.

    As the evaporation of the particle, the scattering signal deviates from a Gaussian distribution, making it inappropriate to directly use the scattering amplitude to calculate $D_\mathrm{p}$. To properly size these particles, the LEO fitting method is employed to reconstruct the Gaussian signal. As described in Sect. 3.3, the zero-crossing point in the TEAPD signal can serve as a position reference

for particles in the SP2. Moreover, the position difference between the zero-crossing point and the peak laser intensity remains constant during measurements. The width of the laser intensity distribution and the position of peak laser intensity relative to the zero-crossing point, both determined by Gaussian fitting of numerous unsaturated pure scattering particles, are used to

constrain the LEO fitting, leaving the fitting amplitude as the only free parameter. Using leading-edge data from the signal onset to 5% of the maximum laser intensity for LEO fitting, can obtain the reconstructed scattering amplitude and further
convert it to particle scattering cross-section. The $D_p$ of internally mixed BC can be derived by inputting the LEO-fitted scattering cross-section, BC core diameter, and the corresponding refractive indices of the core and coating into the Mie calculation model (Laborde et al., 2012; Liu et al., 2014; Schwarz et al., 2008; Taylor et al., 2015).

Based on the particle size calculations, the optical properties of particles can be further derived. For pure scattering particles
and externally mixed BC, Mie theory can be used to calculate the $C_{sca}$ and $C_{abs}$ with known refractive index and optical size of particles. For internally mixed BC, a core-shell model is applied, considering the particle as having an ideal BC core surrounded by a uniform non-absorbing coating material. The Mie algorithm of core-shell structured particles is used to obtain the optical cross-section of internally mixed BC. In these calculations, a complex refractive index of $1.95 + 0.96i$ is used for the BC core (Moteki et al., 2023), while a complex refractive index of $1.5 + 0i$ is applied for the coating of internally mixed BC and pure
scattering particles (Schnaiter et al., 2005).

Considering the low signal-to-noise ratio for small particles in SP2, this study applies specific size limits for different particle types. For the pure scattering particles, the smallest size limit for $D_p$ is set at 170 nm (Schwarz et al., 2006; Sedlacek et al., 2015). And for BC-containing particles, the smallest size limits for $D_c$ and $D_p$ are set as 90 nm and 120 nm, respectively.
Additionally, the largest size limit of 600 nm is applied to all particle size. In addition to these size limits, particles that produce signals exceeding the SP2 detection threshold are also excluded from the ML dataset due to incomplete signal recordings. Although the original particle sizes for these saturated signals can often be estimated through the LEO fitting method, these particles are not included in the ML dataset to ensure data quality.

## 3.5 LightGBM model

LightGBM is a novel GBDT (Gradient Boosting Decision Tree) algorithm. In resemblance to GBDT, the objective output of each tree is determined by the discrepancy between the prediction of the tree model and the expected output from the preceding tree, while the input features remain unchanged. The final prediction is generated through an ensemble of decision trees. Different from traditional GBDT algorithms, LightGBM employs a histogram-based algorithm to avoid calculating all continuous features and takes discrete bins as the unit, which consumes less memory and reduces the complexity of data
separation to speed up the training process (Fan et al., 2019). Additionally, LightGBM implements a leaf-wise strategy to grow trees, identifying the leaf with the maximum gain in split variance to perform the split, which is greedier than the traditional level-wise strategy (Gan et al., 2021). Furthermore, LightGBM incorporates two key optimization techniques: gradient-based one-sided sampling and exclusive feature bundling (Sun et al., 2020). These innovations effectively address the challenges associated with processing large-scale datasets and high-dimensional feature spaces, respectively, enhancing the algorithm's
scalability and computational efficiency while maintaining predictive accuracy.

The development of an effective LightGBM model requires careful optimization of multiple hyperparameters, which significantly influence the model's performance across different applications. Table 1 lists the hyperparameters adjusted in this study and their corresponding descriptions. In this study, the SP2 data collected from May 11 to May 25, 2022, is used for establishing the model. The dataset is randomly partitioned into training and testing sets with a ratio of 7:3, as shown in Fig.

S2. The training set is used to train the LightGBM regression model, while the testing set is used to evaluate the accuracy of the model. The GridSearchCV with 5-fold cross-validation is employed to optimize the hyperparameter configuration of the LightGBM inversion model. This comprehensive approach facilitates an exhaustive search for the optimal hyperparameter combination within a predefined parameter space (Ahmad et al., 2022). By utilizing cross-validation, the methodology effectively mitigates the risk of overfitting and provides robust estimates of the model's generalization performance.

Additionally, an early stopping mechanism is integrated to further prevent potential model overfitting. All optimized hyperparameters are presented in Table 2.

**Table 1.** The main hyperparameters of the LightGBM model tuned in this study.

| Hyperparameters | Description |
|---|---|
| learning_rate | Control the shrinkage rate. |
| num_leaves | Control the maximum number of leaves of a decision tree. |
| max_bin | Control the max number of bins (data intervals) when the dataset of a parameter in the input layer is transformed to a histogram. |
| max_depth | Limit the max depth for a tree model. |
| feature_fraction | The proportion of the selected parameters to the total number of the parameters in the input layer. |
| bagging_fraction | The proportion of the selected data to the total data size. |
| bagging_freq | The frequency of re-sampling the data when bagging_fraction is smaller than 1.0. |

**Table 2.** The optimal hyperparameters for each particle type. The content in parentheses following the particle type name indicates the physical quantity that needs to be inverted for that type of particle.

| Hyperparameters | Particle Type | | | |
|---|---|---|---|---|
| | Pure scattering particle ($D_p$ / $C_{sca}$) | Externally mixed BC ($C_{sca}$ / $C_{abs}$) | Internally mixed BC ($D_p$ / $C_{sca}$ / $C_{abs}$) | BC-containing particle ($D_c$) |
| learning_rate | 0.1 | 0.1 | 0.05 | 0.05 |
| num_leaves | 50 | 50 | 700 | 45 |
| max_bin | 800 | 1000 | 500 | 800 |
| max_depth | 15 | 15 | 50 | 15 |
| feature_fraction | 0.9 | 0.8 | 0.7 | 0.9 |
| bagging_fraction | 0.9 | 0.9 | 0.8 | 0.9 |
| bagging_freq | 2 | 3 | 4 | 2 |

### 3.6 Model performance evaluation

A comprehensive evaluation of the BC mixing state inversion model is essential for validating its reliability. The evaluation metrics employed in this study include the coefficient of determination $R^2$, root mean square error (RMSE), and mean absolute error (MAE). The $R^2$ metric, ranging from 0 to 1, serve as the most important indicator of model accuracy, where a value of 1 indicates perfect prediction without bias. In general, higher $R^2$ indicate better model performance. RMSE quantifies the standard deviation of the residuals between the predicted value and observed value calculated as:

$$\text{RMSE} = \sqrt{\frac{1}{m}\left(\sum_{i=1}^{m}(y_i - \hat{y}_i)^2\right)}\,, \tag{1}$$

where $m$ is the number of samples, $y_i$ is the actual value and $\hat{y}_i$ is the predicted value of $i^{th}$ sample. MAE is another statistical measure to evaluate the bias between predicted value and observed value, which is defined as:

$$\text{MAE} = \frac{1}{m}|y_i - \hat{y}_i|\,. \tag{2}$$

In general, the lower RMSE and MAE values represent the better fitting results of the model.

### 3.7 Model explanation

The increasing complexity of machine learning models often creates a "black box" effect, making it challenging to interpret how input parameters influence prediction results. To address this limitation and enhance model transparency, we implement the SHAP method, which provides insights into the model's decision-making process.

SHAP is a state-of-the-art approach to model interpretation, combining optimal credit allocation with local explanations using Shapley values from game theory (Lundberg and Lee, 2017). It can be used in conjunction with different ML models for model interpretation. Tree-SHAP (Lundberg et al., 2019) is specifically employed in this study, which utilizes a linear explanatory model and Shapley values to estimate the original prediction model, as defined by Eq. (3):

$$f(x) = \Phi_0 + \sum_{i=1}^{p}\Phi_i\,, \tag{3}$$

where $f(x)$ represents the machine learning model's prediction; $\Phi_0$ is the base value of the model, which denotes the average prediction of all inputs; $\Phi_i$ is the SHAP value for feature $i$, indicating the contribution of feature $i$ to the prediction; and $p$ is the total number of features. The SHAP framework provides a unified measure of feature importance, enabling detailed analysis of how individual features influence the model's predictions.

# 4 Result

## 4.1 Inversion results of particle size

The model performance is evaluated using testing data. Figure 4 illustrates the particle size inversion results for three different particle types. The LightGBM model demonstrates excellent capability in predicting particle sizes. The $D_c$ inversion of BC-containing particles shows superior performance (Fig. 4b) with an $R^2$ value of 0.99. Additionally, RMSE and MAE values are 0.36 nm and 0.14 nm, respectively, which are the smallest among the three particle size inversion models, can be attributed to the linear correlation between the peak intensity of incandescence signals and the mass of rBC. The minor discrepancies observed between the predictions and measurements for BC-containing particles with large $D_c$ can be attributed to limited representation of these larger particles in the training dataset, which constrains the model's learning capability in this size range. The $D_p$ inversion for pure scattering particles also exhibits high accuracy (Fig. 4a), achieving an $R^2$ value of 0.99, with RMSE and MAE values of 0.95 nm and 0.51 nm. The slightly lower accuracy compared to the $D_c$ inversion for BC-containing particles may be due to laser intensity fluctuations caused by instrument voltage variations, which affect the scattering amplitude used for $D_p$ calculation. The significant deviations (> 30 nm) observed in a small number of particles are primarily attributed to the instrument noise that produces abnormal scattering signals.

For internally mixed BC, the $D_p$ inversion model achieves an $R^2$ value of 0.98, with RMSE and MAE values of 5.85 nm and 2.98 nm, respectively (Fig. 4c). The close $R^2$ values for the training and testing sets demonstrate the model's excellent generalization performance (Fig. S3). According to density distribution, the predicted values are closely aligned with the observations for most particles. Compared to the previous two particle types, the relationship between $D_p$ and both scattering and incandescence signals is nonlinear, making the physical inversion process more complex and involving more input variables, which increases the difficulty of inversion.

To comprehensively assess the model's performance across different particle size ranges, we further analyzed the prediction error distribution for $D_p$ inversion model of internally mixed BC, as shown in Fig. 5. For particles smaller than 150 nm, the prediction errors average around 4 nm, primarily due to the low signal-to-noise ratio of their scattering signals, which introduces larger uncertainties in the LEO fitting process. The model exhibits optimal performance for particles between 150 nm and 300 nm, with an average prediction error of approximately 1.5 nm. Furthermore, based on the 25% and 75% percentiles of the error distribution, the model's prediction errors exhibit minimal fluctuation within this size range. However, prediction errors gradually increase with particle size, becoming particularly significant for particles larger than 480 nm. This trend can be attributed to occasional irregular signals at larger sizes, such as scattering or incandescence signals with abnormally broad peak widths. These signal irregularities pose challenges to the accurate characterization of particle physical properties, affecting both LEO fitting accuracy and ML model predictions, potentially leading to more pronounced discrepancies between the two methods. The number size distribution of internally mixed BC in the testing set indicates that most particles fall within

the 150–300 nm range, where the model demonstrates highest accuracy. Although the prediction errors are relatively larger at both ends of the size distribution (< 150 nm and > 400 nm), the number of particles in these ranges is comparatively small, thus having limited impact on the overall performance of the model.

It is worth noting that the LEO fitting method and ML method utilize different parts of original signals, which can lead to discrepancies in $D_p$ values. A detailed discussion of this deviation is presented in Sect. 4.3. Furthermore, the complex mixing structures and morphologies of BC-containing particles also affect the accuracy of both methods (Pang et al., 2022; Wang et al., 2017, 2021a, b). Given that the quantitative impact of these factors is challenging to determine currently, this study does not involve discussion related to this aspect, leaving it as a direction for future research.

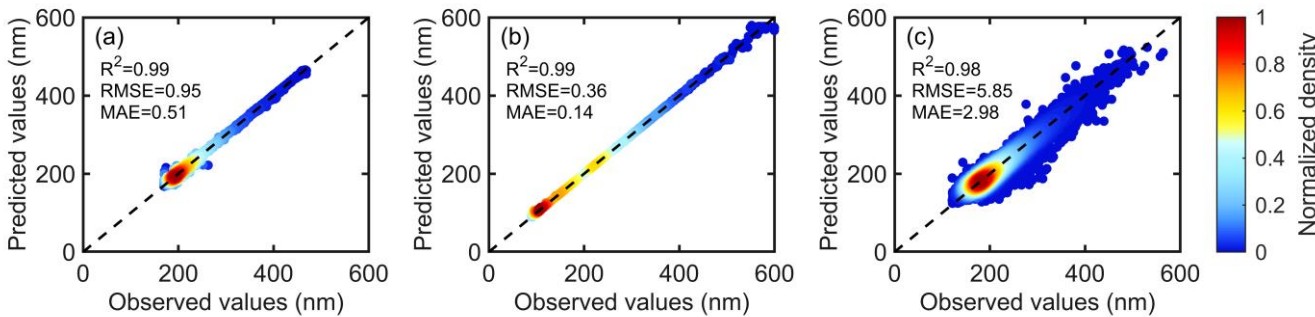


**Figure 4.** Inversion results of particle size for different types of particles: (a) $D_p$ of pure scattering particles; (b) $D_c$ of BC-containing particles; (c) $D_p$ of internally mixed BC.

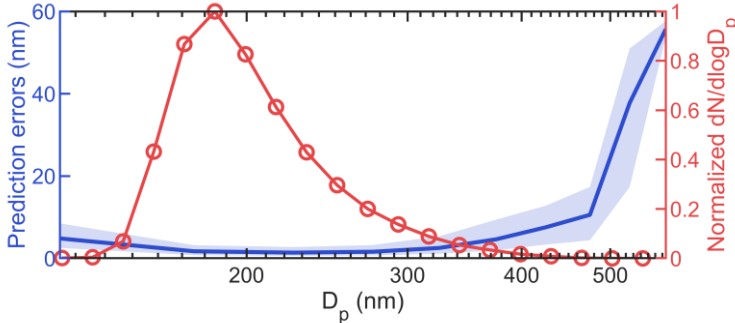

**Figure 5.** The prediction error distribution for $D_p$ inversion model of internally mixed BC, and normalized number size distribution for $D_p$
of internally mixed BC in the testing set. The solid lines in error distribution represent the median value, the upper and lower boundary of the is between the 25 % and 75 % quantiles.

## 4.2 Inversion results of optical properties

The inversion results for the scattering and absorption cross-sections across three particle types are shown in Figs. 6 and 7. Overall, the LightGBM model shows excellent performance in predicting both $C_{sca}$ and $C_{abs}$ for all particle types, with $R^2$

exceeding 0.98. For pure scattering particles, the $C_{sca}$ inversion model exhibits particularly high accuracy with an R² value of 0.99 (Fig. 6a), indicating strong agreement between model predictions and observed values. Similarly, for externally mixed BC, both $C_{sca}$ and $C_{abs}$ inversion models achieve remarkable accuracy with R² values of 0.99 (Figs. 6b and 7a). For internally mixed BC (Figs. 6c and 7b), the model achieves R² values of 0.98 and 0.99 for $C_{sca}$ and $C_{abs}$ inversions, respectively, with $C_{abs}$ predictions showing slightly superior performance. While a small number of particles exhibit notable deviations, the model

predictions demonstrate strong consistency with observed values for majority of particles.

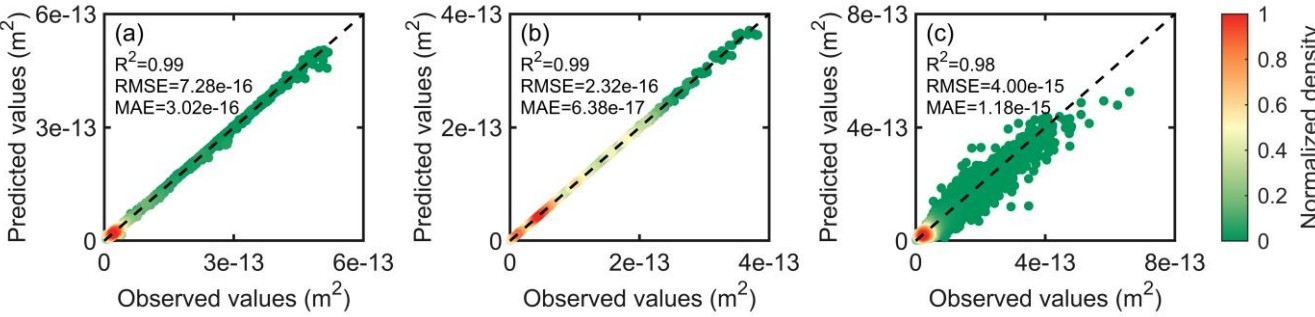

**Figure 6.** Inversion results of $C_{sca}$ for three types of particles: (a) pure scattering particles; (b) externally mixed BC; (c) internally mixed BC.

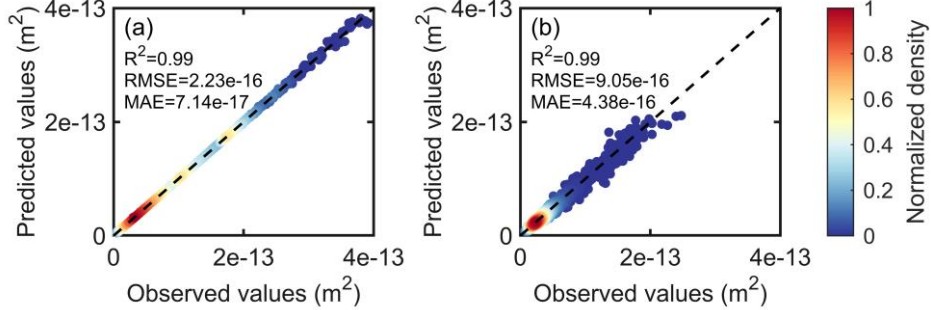

**Figure 7.** Inversion results of $C_{abs}$ for externally mixed BC (a) and internally mixed BC (b).

**4.3 SHAP interpretations**

To understand the relative contributions of input features to the model predictions, the SHAP method is employed for feature importance analysis. Figure 8 presents the analysis results, where individual data points from the dataset are represented with colors indicating their corresponding feature values (blue to red for increasing values). The features are ranked by importance along the y-axis, with higher positions denoting greater significance. The position of each point on the x-axis represents its

SHAP value, quantifying the impact of the feature value corresponding to that point on the model prediction. Points on the right side of the zero line indicate a positive contribution to the model prediction, whereas those on the left side indicate a negative effect. The magnitude of impact is represented by the distance from the zero line. For clarity, the 45-dimensional

scattering signals of the input model are designated as SCLG1, SCLG2, ..., SCLG45, and the 45-dimensional incandescent signals are named similarly as BBLG1, BBLG2, ..., BBLG45.


The SHAP summary plot for the $D_p$ inversion model of internally mixed BC highlights the top fifteen contributing features (Fig. 8a), comprising eight scattering signal features and seven incandescence signal features. The specific distribution of these features within the signals is shown in Fig. 8b. According to the SHAP summary plot, the top three important features are SCLG22, SCLG23, and SCLG21, which correspond to three consecutive scattering signal positions. Similarly, the features

SCLG15, SCLG14, and SCLG16 ranked 7th, 11th, and 12th respectively form another continuous segment of the scattering signal. These six features, located near the peak of the scattering signal, show a positive correlation between their values and predicted $D_p$, as evidenced by their SHAP values. This part of the signal represents a non-linear combination of coating evaporation and incident laser intensity changes. Although this portion of the scattering signal deviates from the original Gaussian profile, it still correlates with the characteristics of the original particle. The intensity of the scattering signal is

proportional to the particle's scattering cross-section, more pronounced signals indicate a larger scattering cross-section, and consequently, a larger $D_p$ value. Among the remaining scattering features, SCLG11 exhibits similar behavior, while SCLG1 shows variable influence (both positive and negative) due to its position at the signal baseline, where instrument noise significantly affects the signal quality.

The seven incandescence signal features (BBLG24 to BBLG30) represent a continuous segment spanning from the onset of the incandescence signal to near-peak position. Their SHAP values all exhibit a trend where their contribution to the prediction transitions from positive to negative as the feature values increase. This behavior reflects potential physical relationships. The incandescence signal peak height exhibits a positive correlation with $D_c$. Under equivalent scattering cross-sections, given that the BC core has a higher scattering coefficient than the coating material, particles with larger $D_c$ values require thinner coating

thickness to maintain the same scattering property, resulting in smaller $D_p$. Additionally, for internally mixed BC with identical $D_c$, while the peak height of the incandescence signal remains constant, the timing of peak occurrence varies with the coating thickness. When the coating is thicker, it takes longer to evaporate, resulting in a delayed onset of the incandescence signal. Consequently, more data points from BBLG24 to BBLG30 are concentrated near the baseline of the incandescence signal, resulting in lower corresponding values (Fig. S4).

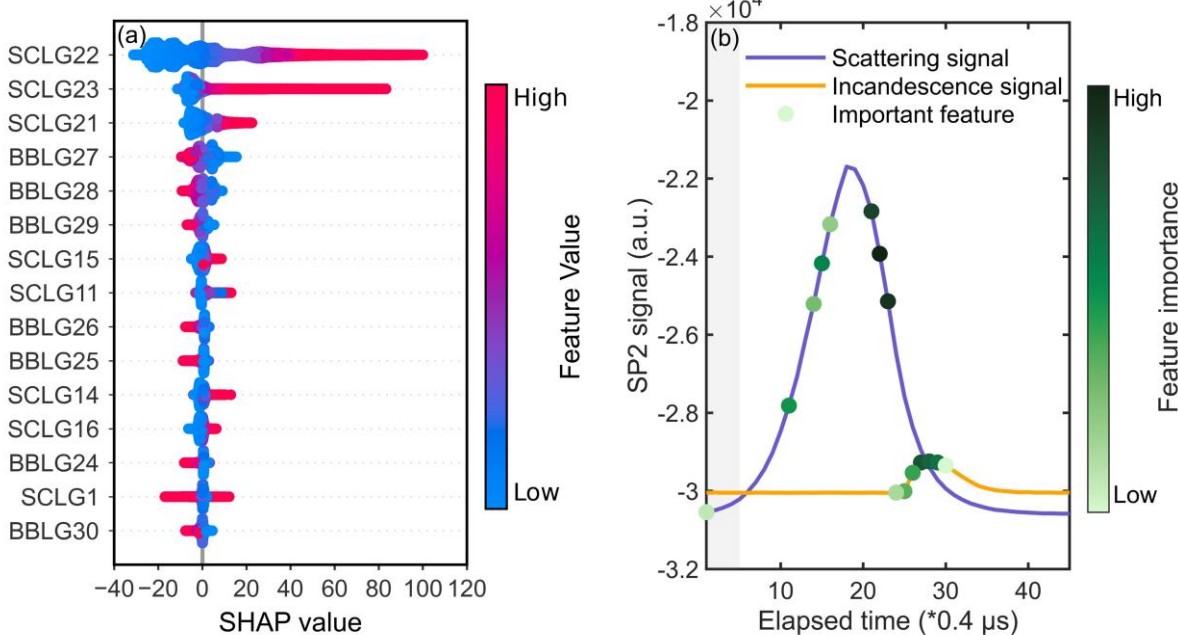


**Figure 8.** SHAP analysis for the $D_p$ inversion model of internally mixed BC. (a) The SHAP summary plot showing the top fifteen features ranked by importance. (b) The specific positions of the top fifteen important features within the scattering and incandescence signals input to the ML model. Scatter points are colored based on their importance ranking (darker colors indicate higher importance). The gray shaded area represents the portion of the scattering signal used for LEO fitting, which is included in the ML model input features.

According to the contribution of each feature indicated by SHAP values, it can be observed that the important features in the ML model differ from the leading-edge data used during the physical inversion process. ML model focuses on the signal near the peak, as shown in Fig. 8b, while the LEO fitting method utilizes signals observed as the BC-containing particle first encounters the laser edge, before coating evaporation occurs. Figure 9 illustrates the LEO fitting results for two different BC-containing particles. Despite nearly identical leading-edge data resulting in similar Gaussian distributions and consequently

the same $D_p$ values through LEO fitting, the complete scattering signals of these particles exhibit significant differences. The ML model, by incorporating these distinctive signal features, can effectively capture these variations, leading to different $D_p$ predictions. Moreover, the leading edge is traditionally defined as the signal from baseline-subtracted zero up to 5 % of the maximum laser intensity (Taylor et al., 2015). As shown in Fig. 9, this portion of the signal (in the grey-shaded area) is close to the baseline, making it particularly susceptible to noise interference. Compared to LEO fitting method, the ML model

utilized a broad range of signals with a high signal-to-noise ratio, demonstrating enhanced noise resistance.

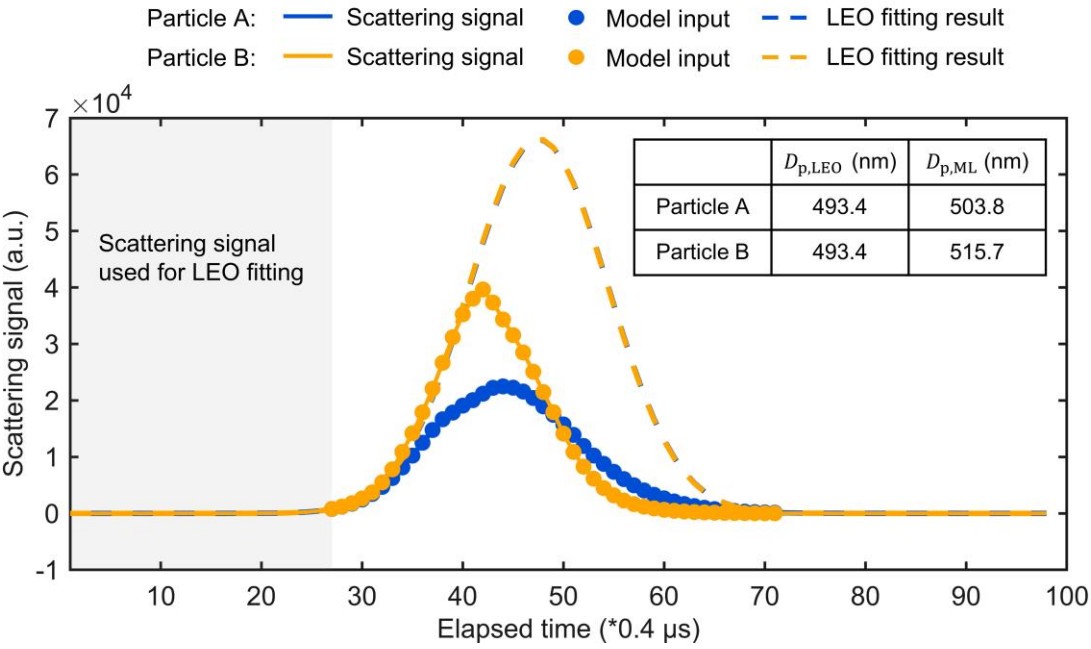

**Figure 9.** Comparison of the scattering signal used in the $D_p$ inversion process for internally mixed BC and corresponding calculation results from both the LEO fitting and the ML methods. The solid line represents the scattering signal obtained by SP2, and the part marked with solid dots is the scattering signal input to the ML model. The gray shaded area shows the leading-edge data used in the LEO fitting process, and the dashed line represents the scattering signal of the original particle reconstructed by LEO fitting.

The SHAP analysis results for the optical cross-section inversion models of internally mixed BC are detailed in Sect. S4.2 of the Supplement. For other particle types, the relationship between target physical properties and feature signals follows relatively straightforward patterns. This study focuses on the more complex case of internally mixed BC, and does not elaborate on the SHAP results of these relatively simple scenarios.

**4.4 Model application**

The BC mixing state inversion model developed in this study demonstrates broad applicability and can be effectively applied to SP2 data obtained across different observation periods. To validate the model's performance, we applied it to the SP2 dataset from April 2022. The results confirm that the model can rapidly and accurately invert the single-particle size information of BC-containing particles. Specifically, the model achieved an R² value of 0.99 for $D_c$ inversion and 0.98 for $D_p$ inversion (Table S1). Based on the inversion results, we analyze the overall size distribution of internally mixed BC in April. The number size distributions shown in Fig. 10 reveal that $D_c$ is predominantly concentrated around 130 nm, while $D_p$ is mainly distributed around 185 nm. The particle size distribution of particles in the two-dimensional histogram indicates that for particles with relatively small $D_c$ (< 100 nm), $D_p$ is primarily distributed around 180 nm. When $D_c$ is small, some particles with thin coatings are not detected due to the detection limits of the SP2, resulting in an overall thicker coating in the SP2 measurement. As $D_c$

gradually increases to the range of 120 to 140 nm, the primary distribution of $D_p$ shifts to around 165 nm, corresponding to a reduction in coating thickness.

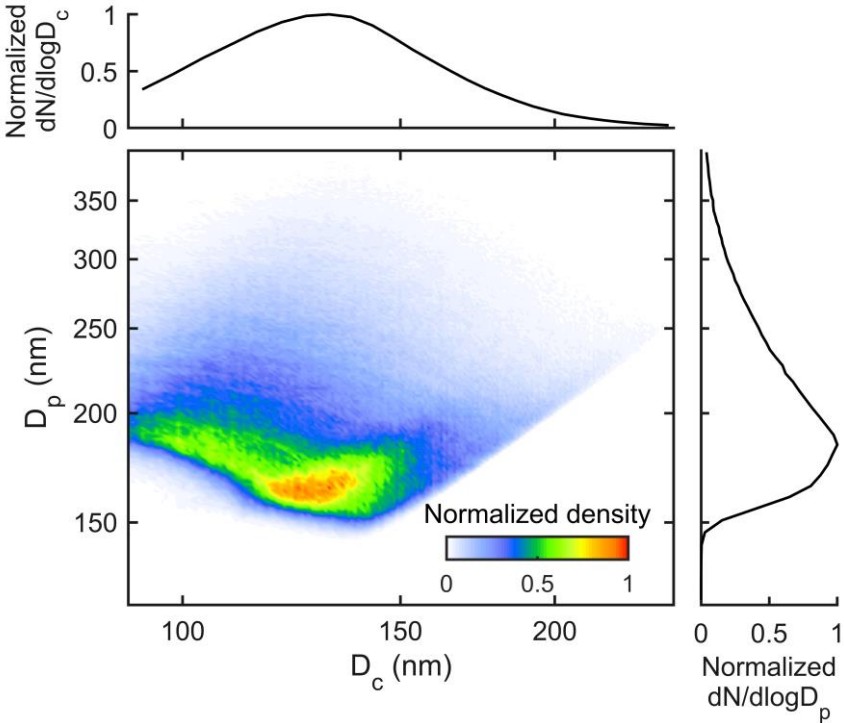

**Figure 10.** Distribution of $D_p$ and $D_c$ of internally mixed BC retrieved from the BC mixing state inversion model. The main panel is a two-dimensional histogram where the color represents the normalized number of particles within a specific size range. Side panels display the
normalized number size distributions of $D_c$ and $D_p$, each scaled to its peak value. Both $D_p$ and $D_c$ axes use logarithmic scales.

Furthermore, comprehensive statistical analysis of BC-containing particles can be derived by combining the BC particle size results with SP2 sampling data. Figure 11 presents the statistical analysis of various physical properties of BC-containing particles in April 2022. The rBC mass concentration varies between 0.24 and 0.36 µg m$^{-3}$ (Fig. 11a), while the relative number fraction of BC-containing particles to the total number of particles ranges from 0.26 to 0.36, with a mean value of 0.31 (Fig.
11b). Both parameters exhibit similar diurnal patterns, showing minimum values in the afternoon due to reduced emissions and the development of the planetary boundary layer (PBL) in the daytime, while maintaining elevated levels throughout the evening. The formation of the nocturnal boundary layer facilitates the accumulation of pollutants, leading to increased rBC mass concentrations during the nighttime and early morning hours (Zhang et al., 2020). During busy traffic periods in the morning and evening, rBC mass concentration and the relative abundance of BC-containing particles increase significantly
due to traffic emissions. Figure 11c shows the diurnal variation of coating thickness (calculated as $D_p-D_c$) of internally mixed BC, with an average value of 78 nm. While the coating thickness remains relatively stable throughout the day, it exhibits greater variability (as indicated by the shaded area in Fig. 11c) during afternoon hours, potentially due to the enhanced BC aging under favorable photochemical conditions. There is a pronounced decrease in the coating thickness of internally mixed

BC at 10:00 LT, which can be attributed to fresh BC emissions from morning traffic. As this newly emitted BC undergoes
initial aging processes and atmospheric mixing, some particles transition from external to internal mixing states. Due to the
brief aging period, these newly internally mixed BC particles exhibit thinner coatings, contributing to a reduction in the mean
coating thickness of the BC population. After 21:00 LT, the coating thickness gradually increases resulting from the nighttime
aging process. Figures. 11d and 11e present the diurnal variations of $D_p/D_c$ and $D_c$ values for internally mixed BC, which serve
as important physical parameters for analyzing BC aging processes. Both parameters show minor fluctuations over the 24
hours. The average $D_p/D_c$ is approximately 1.63, with its diurnal variation pattern closely following that of the coating thickness.
The mean $D_c$ value is around 136 nm, showing a slight increase during afternoon hours.

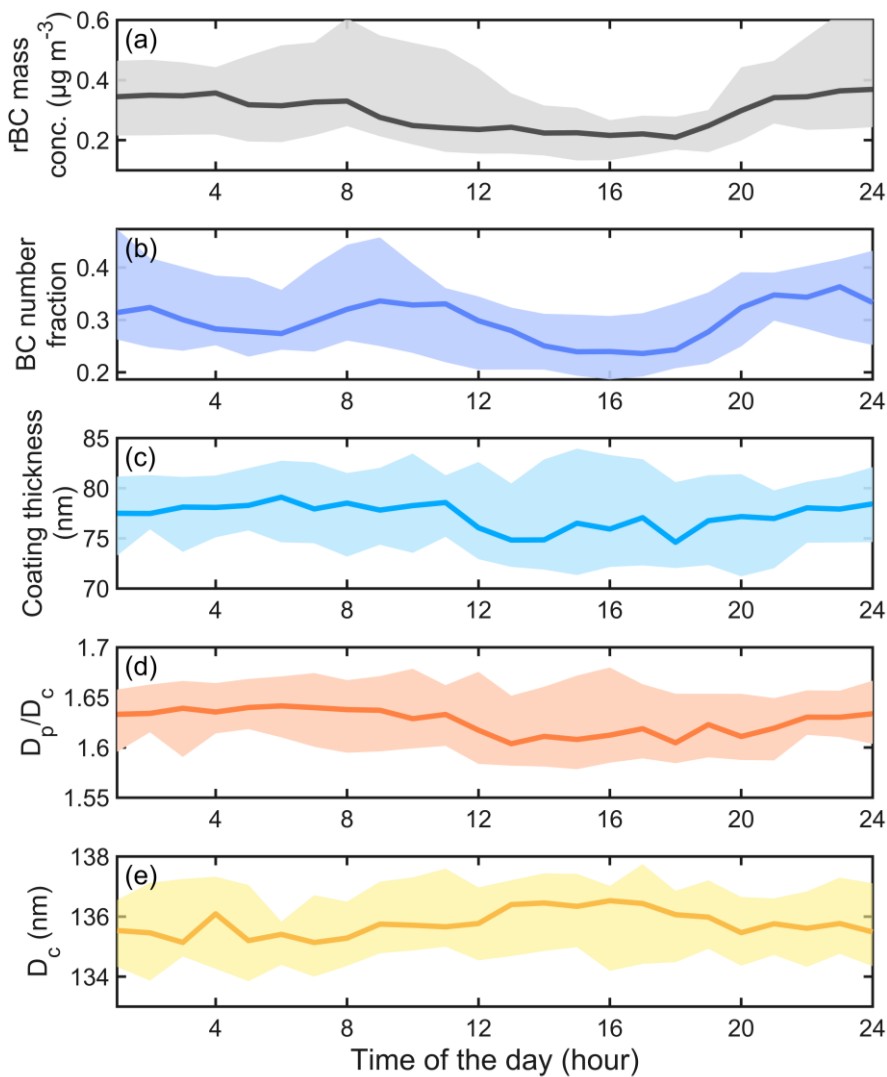

**Figure 11.** The diurnal cycles of (a) the rBC mass concentration; (b) the relative number fraction of BC-containing particles to the total number of particles; (c–e) the coating thickness, $D_p/D_c$, and $D_c$ values of internally mixed BC. The solid lines represent the median value. The upper and lower boundary of the shaded area is between the 25% and 75% quantiles.

**5 Conclusion**

This study conducted comprehensive explorations on the relationship between SP2 data and BC mixing state, developing a ML-based inversion model using LightGBM to correlate SP2 signals with particle size and optical properties. Our results demonstrate that the inversion model can efficiently retrieve the size of various particle types along with their optical properties. The $R^2$ between the predicted and observed values can reach 0.98 or higher. This model can serve as an effective alternative to traditional physical inversion methods, simplifying the quantitative analysis process of particle size and optical properties.

Further, we employed the SHAP method to evaluate the relative importance of SP2 signal features in the $D_p$ inversion model for internally mixed BC and investigated their underlying physical mechanisms. Compared to the LEO fitting method, the ML method utilizes a broader range of signals, including the scattering signal peak rather than solely relying on leading-edge data. This comprehensive signal utilization enables a more accurate characterization of the diverse particle properties. Moreover, the ML method uses signals with a high signal-to-noise ratio, providing better noise resistance. The LightGBM algorithm further strengthens the model's robustness by determining output values through averaging samples within leaf nodes.

Based on our developed model, we can extract statistical characteristics of BC-containing particles, including rBC mass concentration, coating thickness of internally mixed BC, BC number fraction, etc. These characteristics provide valuable insights into the physical properties of BC particles. We validated the model's effectiveness and applicability using the SP2 dataset from the SORPES station. The results confirm that our model can rapidly and accurately derive various physical properties of BC-containing particles. Analysis revealed that the BC number fraction ranges from 0.26 to 0.36, with a mean value of 0.31. The diurnal variations in coating thickness and $D_p/D_c$ of internally mixed BC exhibit relatively stable patterns, with average values of 78 nm and 1.63, respectively. With this model, online real-time mixing state analysis of single-particle measurement is realized. Additionally, given its simplicity and practicality, this approach holds significant potential for wide application in environmental monitoring and climate studies.

**Code and data availability.**

The data and codes related to this article are available upon request from the corresponding author.

**Supplement.**

The supplement related to this article is available online at:

**Author contributions.**

JianW and JiapW designed and directed the study. ZT contributed to algorithm development and data analysis and wrote the manuscript. JX provided guidance on machine learning models. JiapW and JinW provided support for data collection. YJ, ZZ, SS, and BW helped modify the grammar of the manuscript. JianW, JiapW, CL, WN, XH, and AD contributed to the data interpretation and review of the manuscript.

**Competing interests.**

The authors declare that none of the authors has no conflict of interest.

**Disclaimer.**

Publisher's note: Copernicus Publications remains neutral with regard to jurisdictional claims made in the text, published maps, institutional affiliations, or any other geographical representation in this paper. While Copernicus Publications makes every effort to include appropriate place names, the final responsibility lies with the authors.

**Acknowledgements.**

We acknowledge the High Performance Computing Center of Nanjing University of Information Science & Technology for their support of this work.

**Financial support.**

This work was supported by the National Natural Science Foundation of China 42075098 (Jiandong Wang) and the National Key R&D Program of China (2022YFC3701000, Task 5, Jiandong Wang).

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
