# Peer review of "Inversion Algorithm of Black Carbon Mixing State Based on Machine Learning"

_EGUsphere, 2024_

## Referee Comment (RC1)

This manuscript presents an innovative and practical approach to deriving black carbon (BC) mixing states using a machine learning (ML) framework, specifically LightGBM. The integration of SHAP analysis for interpretability and the application of the model to real-world data significantly enhance its relevance. The study is methodologically sound, with comprehensive results demonstrating the model's robustness and applicability. With previous comments addressed, further minor clarifications, particularly in defining particle categories and expanding on error analysis, will further enhance the paper's clarity and scientific rigor. Overall, this is a strong contribution to the field and can be published after minor revisions.

**Specific comments:**

1. Line 15-20: Briefly explain why LightGBM was chosen over other models like Random Forest or Neural Networks. Highlight its advantages for handling large datasets or nonlinear relationships.

2. Expand the discussion on how this method improves upon or complements existing techniques, such as the LEO fitting method or other machine learning approaches. For example, what specific challenges of previous methods (e.g., noise resistance, scalability) does this model overcome?

3. Although the study uses data from a single site, could the authors discuss the expected performance of the model in different environmental conditions or geographical regions? For example, how might differences in aerosol composition affect results?

4. Consider adding uncertainty quantification for model predictions. For instance, providing confidence intervals for particle size or optical property predictions would help assess reliability in practical applications.

5. The SHAP analysis identifies important features in the scattering and incandescence signals. However, the physical relevance of these features (e.g., why certain regions of the signal are more predictive) could be discussed more thoroughly.

---

## Referee Comment (RC2)

**Inversion Algorithm of Black Carbon Mixing State Based on Machine Learning**

The mixing states of black carbon are widely measured by the single-particle soot photometer (SP2) instrument. This study employed a machine learning (ML) based method, light gradient boosting machine (LightGBM), to process the scattering and incandescence signals of SP2 and to retrieve the mixing states of particles. ML based method performs more efficiently than the traditional Leading-Edge-Only (LEO) approach with quite consistent retrieval outcomes. The relative importances of the selected signal features in retrieving the particle microphysical properties were studied by SHapley Additive exPlanation (SHAP) method. The authors stated that this ML based method has the potential to be a reliable noise-resistant approach to analyze the SP2 data. My major comments are attached below:

**Main comments:**

1. Fig.3, 4, & 5 compare the predicted particle microphysical properties with those named "actual values". How did you define the "actual values", and how they were obtained? Were they the outputs from LEO approach or did you use any other particle sizer instruments to measure the "actual values" of particle size? According to the meaning of "actual", this value should be regarded as the ground-truth, or more reliable measurements of the particle size.

2. Fig. 7 shows the robustness of the ML-based retrievals by comparing the retrieved particle size with the one obtained by LEO approach. LEO method, because it utilizes part of the scattering signal, has the possibility to mischaracterize the particles with different sizes. Comments: I agree that the retrieval accuracy will be improved with more observational constraints or signal feature inputs. However, the reason that LEO method only utilizes the threshold portion but not the entire scattering signals is that the loss or vaporization of particle coatings happens after the particle started to absorb laser energy in SP2. The entire scattering signal function doesn't reflect the scattering properties of the original mixing states of BC-containing particles (coating thickness, BC core size, etc.). Therefore, LEO method utilizes the threshold portion of the signal when the particle properties (size) doesn't change significantly yet in SP2. Though the proposed ML-based method utilizes more signal feature, it doesn't necessarily reflect the true size of the original coated particles.

3. Table 2: This table shows the hyperparameters for each particle type. What are the "Internally mixed BC" and "BC-containing particle" by definition? BC-containing particle is a subset of internally mixed BC in my opinion. Then why did you use different hyperparameters for them?

**Line-by-line comments:**

Line 73: "quality of the refractory components", what does "quality" mean here?

Line 76: Add reference here (Gao, R. S., et al., 2007, Aerosol Science & Technology)

Line 124: "45-dimensional scattering signals". What is the relationship between the 45-dimensional signals and the abovementioned "100-dimensional signals of SP2 data" in line 94, and 90-dimenional feature data in line 137. It would be better to provide additional contexts/descriptions of the principles of signal feature selection for optical retrievals.

Line 191: What is "GridSearchCV function"

Line 191: Please check the grammar of the sentence starting with "Based on". What is the subject of this sentence?

---

## Author Comment (AC1)

**Response to the comments of Reviewer #1 (EGUSPHERE-2024-2496)**

*The mixing states of black carbon are widely measured by the single-particle soot photometer (SP2) instrument. This study employed a machine learning (ML) based method, light gradient boosting machine (LightGBM), to process the scattering and incandescence signals of SP2 and to retrieve the mixing states of particles. ML based method performs more efficiently than the traditional Leading-Edge-Only (LEO) approach with quite consistent retrieval outcomes. The relative importances of the selected signal features in retrieving the particle microphysical properties were studied by SHapley Additive exPlanation (SHAP) method. The authors stated that this ML based method has the potential to be a reliable noise-resistant approach to analyze the SP2 data. My major comments are attached below.*

**Response:** We sincerely thank the reviewer #1 for the insightful suggestions and constructive comments. In response, we have thoroughly revised the manuscript and prepared a detailed, point-by-point response to all comments and questions raised. In the revised manuscript, we have added conceptual definitions for different particle types and clarified their classification criteria. Additionally, we provide a more detailed description of the feature dataset construction process, explaining the intrinsic connection between the feature signals used in the model and the original signals, as well as their scientific significance. Here are our point-to-point responses.

**Main Comments:**

*1) Fig.3, 4, & 5 compare the predicted particle microphysical properties with those named "actual values". How did you define the "actual values", and how they were obtained? Were they the outputs from LEO approach or did you use any other particle sizer instruments to measure the "actual values" of particle size? According to the meaning of "actual", this value should be regarded as the ground-truth, or more reliable measurements of the particle size.*

**Response:** Thank you for pointing this out. These particle microphysical properties named "actual values" are derived through physical inversion methods, as detailed in the "Construction of label dataset" section of our manuscript. Specifically, for internally mixed BC, these values are outputs from the LEO approach. We employed the machine learning model to learn the mapping relationship between the input (SP2 signals) and output (microphysical properties) data. Based on the developed inversion model, when we input SP2 signals, the model can predict the corresponding microphysical properties. These predicted values are then compared with the "actual values" obtained through the physical inversion methods to evaluate the model's performance, as shown in Fig. 3, 4, & 5.

We appreciate your reminder, and we realize that using the "actual values" in the

manuscript is not appropriate. In response, we have revised it to "observed values".

*2) Fig. 7 shows the robustness of the ML-based retrievals by comparing the retrieved particle size with the one obtained by LEO approach. LEO method, because it utilizes part of the scattering signal, has the possibility to mischaracterize the particles with different sizes. Comments: I agree that the retrieval accuracy will be improved with more observational constraints or signal feature inputs. However, the reason that LEO method only utilizes the threshold portion but not the entire scattering signals is that the loss or vaporization of particle coatings happens after the particle started to absorb laser energy in SP2. The entire scattering signal function doesn't reflect the scattering properties of the original mixing states of BC-containing particles (coating thickness, BC core size, etc.). Therefore, LEO method utilizes the threshold portion of the signal when the particle properties (size) doesn't change significantly yet in SP2. Though the proposed ML-based method utilizes more signal feature, it doesn't necessarily reflect the true size of the original coated particles.*

**Response:** Thank you for your comment. As you described, the LEO fitting uses the leading edge of the scattering signal, which corresponds to the stage before the coating evaporates, aiming to capture the original characteristics of the particle when it has not yet significantly changed in the SP2. While, as we pointed out in the discussion of Fig. 7 (which corresponds to Fig. 9 in the revised manuscript), the leading edge of the scattering signal is close to the baseline and thus more susceptible to noise interference, which may increase uncertainty in the LEO fitting.

As the BC core absorbs laser energy, the coating evaporates upon reaching its boiling point, the scattering signal gradually deviates from a Gaussian distribution. Although this distorted part of the signal cannot directly reflect the initial physical characteristics of the particle, the changes in the signal are not entirely random. This signal can be considered a complex function of the BC core diameter ($D_c$) and the entire particle diameter ($D_p$). In fact, previous studies have tried to use these changes in the scattering signal to depict the variation of particle scattering cross-section in the SP2 or the mixing state of BC (Moteki et al., 2014; Moteki and Kondo, 2008; Sedlacek et al., 2012).

Considering that the leading-edge signal only accounts for a small portion of the entire scattering signal, the information it can provide is limited. Therefore, this study attempts to input the complete scattering signal change into a machine learning model, aiming to parse the information contained in the subsequent signal. This approach can make more comprehensive use of the information expressed by the signal, thereby providing more robust results.

*3) Table 2: This table shows the hyperparameters for each particle type. What are the "Internally mixed BC" and "BC-containing particle" by definition? BC-containing particle is a subset of internally mixed BC in my opinion. Then why did you use different hyperparameters for them?*

**Response:** Thank you for your question. We have provided more detailed explanations of the definitions for different particle types in the revised manuscript. Lines 109 to 122 in the revised manuscript reflect the specific revisions:

*"In this study, ambient particles measured by SP2 are classified into pure scattering particles and BC-containing particles. **Pure scattering particles are those that only scatter light without significant absorption, while BC-containing particles, which contain refractory BC (rBC), both scatter and absorb light. BC-containing particles are further subdivided into externally mixed BC and internally mixed BC. Externally mixed BC refers to freshly emitted BC particles that have not yet mixed with other aerosol components, while internally mixed BC describes BC that has undergone atmospheric aging processes and incorporated with other materials (Oshima et al., 2009).** Operationally, we differentiate the pure scattering particle and BC-containing particle depending on whether it has the incandescence signal. ... **The incandescence signal peak occurs when all non-BC material has evaporated and the BC reaches its incandescence temperature, thus the magnitude of $\Delta t$ correlates with the thickness of the coating on BC particles: a larger $\Delta t$ corresponds to a thicker coating that takes longer to evaporate. By examining the distribution of $\Delta t$ values in the SP2 measurements, as illustrated in Fig. 2d (Sedlacek et al., 2012; Subramanian et al., 2010; Zhang et al., 2016)**, BC-containing particles with $\Delta t < 2\,\mu s$ are classified as externally mixed BC (Fig. 2b), while those with $\Delta t \geq 2\,\mu s$ are categorized as internally mixed BC (Fig. 2c)."*

In addition, the hyperparameter tuning is optimized for the specific tasks of each model, rather than using a universal setting. By adjusting the hyperparameters, we establish the mapping between the model's inputs and outputs. Since each model uses different features and inverses different microphysical properties, the hyperparameters used in model construction also vary accordingly.

**Line-by-line comments:**

*1) Line 73: "quality of the refractory components", what does "quality" mean here?*

**Response:** Thank you for pointing this out. The "quality" here refers to the mass of the refractory BC. For clarity, we have replaced it with the word "mass" in the revised manuscript.

Line 84 in the revised manuscript:

*"The intensity of this thermal radiation depends on the composition and __mass__ of the refractory components, regardless of the particle morphology and mixing state (Schwarz et al., 2006; Slowik et al., 2007)."*

*2) Line 76: Add reference here (Gao, R. S., et al., 2007, Aerosol Science & Technology)*

**Response:** Thank you for your reminder. We have added the reference in the corresponding position.

Line 88 in the revised manuscript:

*"Particle size, therefore, can be measured based on the amount of light they scatter from the laser, which exhibits a Gaussian dependence with time **(Gao et al., 2007)**."*

*3) Line 124: "45-dimensional scattering signals". What is the relationship between the 45- dimensional signals and the abovementioned "100-dimensional signals of SP2 data" in line 94, and 90-dimenional feature data in line 137. It would be better to provide additional contexts/descriptions of the principles of signal feature selection for optical retrievals.*

**Response:** Thank you for your comment. The SP2 signal is recorded based on the elapsed time, with each time window corresponding to information about a single particle. For each particle, the corresponding original scattering signal and incandescence signal are both 100-dimensional. The position of particles within the instrument is not known in advance. Among SP2's four detectors, there is a two-element APD (TEAPD) detector. This detector has a gap perpendicular to the particle's direction of motion, resulting in a notch in the TEAPD signal, as shown in Fig. R1a. Given the stability of SP2's optical alignment and constant sample flow rate, this notch provides a precise time reference for a particle's position within the instrument. In practice, the signal from the leading element is inverted, transforming the notch into a zero-crossing point (Fig. R1b) (Gao et al., 2007). Since SP2 simultaneously records data from all four detector channels, this time reference is valid for the signals from the other three detectors as well.

[Figure]

**Figure R1.** (a) The original scattering signal measured by TEAPD before the signal from the leading element is inverted. (b) The TEAPD signal obtained by SP2, with the blue asterisk indicating the position of the zero-crossing point.

For pure scattering particles, we locate the zero-crossing point in the scattering signal and then extract 22 data points both before and after it, creating a 45-dimensional feature dataset (Fig. R2a). In this newly constructed 45-dimensional feature signal, particles are positioned at consistent locations within the instrument for each corresponding dimension, eliminating the influence of laser intensity distribution on the scattering signal.

When inverting the $D_c$ of BC-containing particles, since the peak intensity of the incandescence signal is positively correlated with the mass of the refractory BC

component in the particle, the peak of the incandescence signal is selected as a reference point, from which 22 data points are extracted both preceding and following this point (Fig. R2b), yielding a 45-dimensional feature dataset. This method ensures the incandescence signal peaks from different BC-containing particles are positioned at the same dimension within the feature dataset, facilitating direct comparisons between particles. The optical properties of externally mixed BC are also determined by the refractory BC component, so the same feature selection method is adopted.

For internally mixed BC, its size and optical cross-section characteristics are reflected by both the scattering and incandescence signals. We select scattering features for internally mixed BC using the same approach as for pure scattering particles. Considering that the relative relationship between the incandescence and scattering signals in the original signal can reflect particle characteristics, the incandescence signal is selected with 22 data points before and after the zero-crossing point in a similar way (Fig. R2c). The 90-dimensional feature signal composed of the selected 45-dimensional scattering signal and 45-dimensional incandescence signal is used as the input feature for the internally mixed BC inversion model.

[Figure]

**Figure R2.** Relationship between the original SP2 signals (line plots) and the feature signals used in machine learning model construction (scatter plots) for different particle types: (a) pure scattering particles; (b) externally mixed BC; (c) internally mixed BC. The method for selecting feature signals used in inverting the core diameter ($D_c$) of BC-containing particles is identical to that used for externally

mixed BC.

We have provided additional descriptions of the principles of signal feature selection for different particle types as suggested. The Figs. R1 and R2 presented here have also been incorporated into the revised manuscript and Supplement Information to further illustrate these principles. Specifically, Fig. R1 has been added as Fig. 3 in the revised manuscript, while Fig. R2 has been included as Fig. S1 in the Supplement Information.

Lines 141 to 143:

*"**The SP2 signal is recorded based on the elapsed time, with each time window corresponding to information about a single particle. For each particle, the original scattering signal and incandescence signal are both 100-dimensional. The position of particles within the instrument is not known in advance.**"*

Lines 154 to 162:

*"* *As mentioned in Sect. 2.2, one of the four detectors in the SP2 is a split APD detector. This detector has a gap perpendicular to the particle's direction of motion, resulting in a notch in the TEAPD signal, as shown in Fig. 3a. Given the stability of SP2's optical alignment and constant sample flow rate, this notch provides a precise time reference for a particle's position within the instrument. In practice, the signal from leading element is inverted, transforming the notch into a zero-crossing point (Fig. 3b) (Gao et al., 2007).* *Since SP2 simultaneously records data from all four detector channels, this time reference is valid for the signals from the other three detectors as well.* *We locate the zero-crossing point in the scattering signal and then extract 22 data points both before and after it, creating a 45-dimensional feature dataset (Fig. S1a). Through this standardization, the differences in signal intensity can be accurately attributed to the inherent physical properties of the particles.*

Lines 168 to 173:

*Based on this characteristic, the peak of the incandescence signal is selected as a reference point, from which 22 data points are extracted both preceding and following this point (Fig. S1b), yielding a 45-dimensional feature dataset used for inverting the $D_c$ of BC-containing particles. This method ensures the incandescence signal peaks from different BC-containing particles are positioned at the same dimension within the feature dataset, facilitating direct comparisons between particles while preserving comprehensive information about the incandescence process.*

Lines 184 to 189:

*"Compared with other particle types, the internally mixed BC has a more complex structure. ... Simultaneously, considering that the relative relationship between the original incandescence and scattering signals can reflect particle characteristics, such as coating thickness (Moteki and Kondo, 2007; Schwarz et al., 2006; Subramanian et al., 2010), the incandescence signal is selected with 22 data points before and after the zero-crossing point in a similar way (Fig. S1c). The feature extraction process yields a 90-dimensional feature dataset, comprising 45-dimensional scattering signal and 45-dimensional incandescence signal, ensuring that we can*

*comprehensively capture the key characteristics of internally mixed BC."*

*4) Line 191: What is "GridSearchCV function".*

**Response:** Thank you for your question. GridSearchCV is an automated hyperparameter tuning technique that combines grid search with cross-validation (Ahmad et al., 2022). In this process, a range of values is defined for each hyperparameter, and then all possible combinations are systematically tried. It uses K-fold cross-validation to evaluate the performance of each hyperparameter combination, eliminating the need for manual adjustments by automatically executing tests for all possible combinations. GridSearchCV assesses performance based on the average results of cross-validation and ultimately selects the hyperparameter combination that produces the best performance. The main advantage of this method lies in its ability to comprehensively and systematically explore the hyperparameter space while providing reliable estimates of the model's generalization capability. We have added a brief explanation of the GridSearchCV function in the revised manuscript.

Lines 251 to 254:

*"The GridSearchCV with 5-fold cross-validation is employed to optimize the hyperparameter configuration of the LightGBM inversion model. **This comprehensive approach facilitates an exhaustive search for the optimal hyperparameter combination within a predefined parameter space (Ahmad et al., 2022). By utilizing cross-validation, the methodology effectively mitigates the risk of overfitting and provides robust estimates of the model's generalization performance.**"*

*5) Line 191: Please check the grammar of the sentence starting with "Based on". What is the subject of this sentence?*

**Response:** Thank you for your careful review and for pointing out this grammatical issue. The sentence beginning with "Based on" lacks a clear subject, which affects its grammatical structure. We have revised the sentence to "The GridSearchCV with 5-fold cross-validation is employed to optimize the hyperparameter configuration of the LightGBM inversion model." [Page 10, Line 251]

**Reference**

Ahmad, G. N., Fatima, H., and Saidi, A. S.: Efficient Medical Diagnosis of Human Heart Diseases Using Machine Learning Techniques With and Without GridSearchCV, 10, 2022.

Gao, R. S., Schwarz, J. P., Kelly, K. K., Fahey, D. W., Watts, L. A., Thompson, T. L., Spackman, J. R., Slowik, J. G., Cross, E. S., Han, J.-H., Davidovits, P., Onasch, T. B., and Worsnop, D. R.: A Novel Method for Estimating Light-Scattering Properties of Soot Aerosols Using a Modified Single-Particle Soot Photometer, Aerosol Science and Technology, 41, 125–135, https://doi.org/10.1080/02786820601118398, 2007.

Moteki, N. and Kondo, Y.: Effects of Mixing State on Black Carbon Measurements by Laser-Induced Incandescence, Aerosol Science and Technology, 41, 398–417, https://doi.org/10.1080/02786820701199728, 2007.

Moteki, N. and Kondo, Y.: Method to measure time-dependent scattering cross sections of particles evaporating in a laser beam, Journal of Aerosol Science, 39, 348–364, https://doi.org/10.1016/j.jaerosci.2007.12.002, 2008.

Moteki, N., Kondo, Y., and Adachi, K.: Identification by single-particle soot photometer of black carbon particles attached to other particles: Laboratory experiments and ground observations in Tokyo, JGR Atmospheres, 119, 1031–1043, https://doi.org/10.1002/2013JD020655, 2014.

Oshima, N., Koike, M., Zhang, Y., Kondo, Y., Moteki, N., Takegawa, N., and Miyazaki, Y.: Aging of black carbon in outflow from anthropogenic sources using a mixing state resolved model: Model development and evaluation, J. Geophys. Res., 114, 2008JD010680, https://doi.org/10.1029/2008JD010680, 2009.

Schwarz, J. P., Gao, R. S., Fahey, D. W., Thomson, D. S., Watts, L. A., Wilson, J. C., Reeves, J. M., Darbeheshti, M., Baumgardner, D. G., Kok, G. L., Chung, S. H., Schulz, M., Hendricks, J., Lauer, A., Kärcher, B., Slowik, J. G., Rosenlof, K. H., Thompson, T. L., Langford, A. O., Loewenstein, M., and Aikin, K. C.: Single-particle measurements of midlatitude black carbon and light-scattering aerosols from the boundary layer to the lower stratosphere, J. Geophys. Res., 111, 2006JD007076, https://doi.org/10.1029/2006JD007076, 2006.

Sedlacek, A. J., Lewis, E. R., Kleinman, L., Xu, J., and Zhang, Q.: Determination of and evidence for non-core-shell structure of particles containing black carbon using the Single-Particle Soot Photometer (SP2), Geophysical Research Letters, 39, 2012GL050905, https://doi.org/10.1029/2012GL050905, 2012.

Slowik, J. G., Cross, E. S., Han, J.-H., Davidovits, P., Onasch, T. B., Jayne, J. T., Williams, L. R., Canagaratna, M. R., Worsnop, D. R., Chakrabarty, R. K., Moosmüller, H., Arnott, W. P., Schwarz, J. P., Gao, R.-S., Fahey, D. W., Kok, G. L., and Petzold, A.:

An Inter-Comparison of Instruments Measuring Black Carbon Content of Soot Particles, Aerosol Science and Technology, 41, 295–314, https://doi.org/10.1080/02786820701197078, 2007.

Subramanian, R., Kok, G. L., Baumgardner, D., Clarke, A., Shinozuka, Y., Campos, T. L., Heizer, C. G., Stephens, B. B., de Foy, B., Voss, P. B., and Zaveri, R. A.: Black carbon over Mexico: the effect of atmospheric transport on mixing state, mass absorption cross-section, and BC/CO ratios, Atmos. Chem. Phys., 2010.

Zhang, Y., Zhang, Q., Cheng, Y., Su, H., Kecorius, S., Wang, Z., Wu, Z., Hu, M., Zhu, T., and Wiedensohler, A.: Measuring the morphology and density of internally mixed black carbon with SP2 and VTDMA: new insight into the absorption enhancement of black carbon in the atmosphere, Atmospheric Measurement Techniques, 9, 1833–1843, 2016.

---

## Author Response (AR1)

**Response to the comments of Reviewer #1 (EGUSPHERE-2024-2496)**

*The mixing states of black carbon are widely measured by the single-particle soot photometer (SP2) instrument. This study employed a machine learning (ML) based method, light gradient boosting machine (LightGBM), to process the scattering and incandescence signals of SP2 and to retrieve the mixing states of particles. ML based method performs more efficiently than the traditional Leading-Edge-Only (LEO) approach with quite consistent retrieval outcomes. The relative importances of the selected signal features in retrieving the particle microphysical properties were studied by SHapley Additive exPlanation (SHAP) method. The authors stated that this ML based method has the potential to be a reliable noise-resistant approach to analyze the SP2 data. My major comments are attached below.*

**Response:** We sincerely thank the reviewer #1 for the insightful suggestions and constructive comments. In response, we have thoroughly revised the manuscript and prepared a detailed, point-by-point response to all comments and questions raised. In the revised manuscript, we have added conceptual definitions for different particle types and clarified their classification criteria. Additionally, we provide a more detailed description of the feature dataset construction process, explaining the intrinsic connection between the feature signals used in the model and the original signals, as well as their scientific significance. Here are our point-to-point responses.

**Main Comments:**

*1) Fig.3, 4, & 5 compare the predicted particle microphysical properties with those named "actual values". How did you define the "actual values", and how they were obtained? Were they the outputs from LEO approach or did you use any other particle sizer instruments to measure the "actual values" of particle size? According to the meaning of "actual", this value should be regarded as the ground-truth, or more reliable measurements of the particle size.*

**Response:** Thank you for pointing this out. These particle microphysical properties named "actual values" are derived through physical inversion methods, as detailed in the "Construction of label dataset" section of our manuscript. Specifically, for internally mixed BC, these values are outputs from the LEO approach. We employed the machine learning model to learn the mapping relationship between the input (SP2 signals) and output (microphysical properties) data. Based on the developed inversion model, when we input SP2 signals, the model can predict the corresponding microphysical properties. These predicted values are then compared with the "actual values" obtained through the physical inversion methods to evaluate the model's performance, as shown in Fig. 3, 4, & 5.

We appreciate your reminder, and we realize that using the "actual values" in the

manuscript is not appropriate. In response, we have revised it to "observed values".

*2) Fig. 7 shows the robustness of the ML-based retrievals by comparing the retrieved particle size with the one obtained by LEO approach. LEO method, because it utilizes part of the scattering signal, has the possibility to mischaracterize the particles with different sizes. Comments: I agree that the retrieval accuracy will be improved with more observational constraints or signal feature inputs. However, the reason that LEO method only utilizes the threshold portion but not the entire scattering signals is that the loss or vaporization of particle coatings happens after the particle started to absorb laser energy in SP2. The entire scattering signal function doesn't reflect the scattering properties of the original mixing states of BC-containing particles (coating thickness, BC core size, etc.). Therefore, LEO method utilizes the threshold portion of the signal when the particle properties (size) doesn't change significantly yet in SP2. Though the proposed ML-based method utilizes more signal feature, it doesn't necessarily reflect the true size of the original coated particles.*

**Response:** Thank you for your comment. As you described, the LEO fitting uses the leading edge of the scattering signal, which corresponds to the stage before the coating evaporates, aiming to capture the original characteristics of the particle when it has not yet significantly changed in the SP2. While, as we pointed out in the discussion of Fig. 7 (which corresponds to Fig. 9 in the revised manuscript), the leading edge of the scattering signal is close to the baseline and thus more susceptible to noise interference, which may increase uncertainty in the LEO fitting.

As the BC core absorbs laser energy, the coating evaporates upon reaching its boiling point, the scattering signal gradually deviates from a Gaussian distribution. Although this distorted part of the signal cannot directly reflect the initial physical characteristics of the particle, the changes in the signal are not entirely random. This signal can be considered a complex function of the BC core diameter ($D_c$) and the entire particle diameter ($D_p$). In fact, previous studies have tried to use these changes in the scattering signal to depict the variation of particle scattering cross-section in the SP2 or the mixing state of BC (Moteki et al., 2014; Moteki and Kondo, 2008; Sedlacek et al., 2012).

Considering that the leading-edge signal only accounts for a small portion of the entire scattering signal, the information it can provide is limited. Therefore, this study attempts to input the complete scattering signal change into a machine learning model, aiming to parse the information contained in the subsequent signal. This approach can make more comprehensive use of the information expressed by the signal, thereby providing more robust results.

*3) Table 2: This table shows the hyperparameters for each particle type. What are the "Internally mixed BC" and "BC-containing particle" by definition? BC-containing particle is a subset of internally mixed BC in my opinion. Then why did you use different hyperparameters for them?*

**Response:** Thank you for your question. We have provided more detailed explanations of the definitions for different particle types in the revised manuscript. Lines 109 to 122 in the revised manuscript reflect the specific revisions:

*"In this study, ambient particles measured by SP2 are classified into pure scattering particles and BC-containing particles.* ***Pure scattering particles are those that only scatter light without significant absorption, while BC-containing particles, which contain refractory BC (rBC), both scatter and absorb light. BC-containing particles are further subdivided into externally mixed BC and internally mixed BC. Externally mixed BC refers to freshly emitted BC particles that have not yet mixed with other aerosol components, while internally mixed BC describes BC that has undergone atmospheric aging processes and incorporated with other materials (Oshima et al., 2009).*** *Operationally, we differentiate the pure scattering particle and BC-containing particle depending on whether it has the incandescence signal. ...* ***The incandescence signal peak occurs when all non-BC material has evaporated and the BC reaches its incandescence temperature, thus the magnitude of Δt correlates with the thickness of the coating on BC particles: a larger Δt corresponds to a thicker coating that takes longer to evaporate. By examining the distribution of Δt values in the SP2 measurements, as illustrated in Fig. 2d (Sedlacek et al., 2012; Subramanian et al., 2010; Zhang et al., 2016)****, BC-containing particles with Δt < 2 µs are classified as externally mixed BC (Fig. 2b), while those with Δt ≥ 2 µs are categorized as internally mixed BC (Fig. 2c)."*

In addition, the hyperparameter tuning is optimized for the specific tasks of each model, rather than using a universal setting. By adjusting the hyperparameters, we establish the mapping between the model's inputs and outputs. Since each model uses different features and inverses different microphysical properties, the hyperparameters used in model construction also vary accordingly.

**Line-by-line comments:**

*1) Line 73: "quality of the refractory components", what does "quality" mean here?*

**Response:** Thank you for pointing this out. The "quality" here refers to the mass of the refractory BC. For clarity, we have replaced it with the word "mass" in the revised manuscript.

Line 84 in the revised manuscript:

*"The intensity of this thermal radiation depends on the composition and **mass** of the refractory components, regardless of the particle morphology and mixing state (Schwarz et al., 2006; Slowik et al., 2007)."*

*2) Line 76: Add reference here (Gao, R. S., et al., 2007, Aerosol Science & Technology)*

**Response:** Thank you for your reminder. We have added the reference in the corresponding position.

Line 88 in the revised manuscript:

*"Particle size, therefore, can be measured based on the amount of light they scatter from the laser, which exhibits a Gaussian dependence with time (Gao et al., 2007)."*

*3) Line 124: "45-dimensional scattering signals". What is the relationship between the 45- dimensional signals and the abovementioned "100-dimensional signals of SP2 data" in line 94, and 90-dimenional feature data in line 137. It would be better to provide additional contexts/descriptions of the principles of signal feature selection for optical retrievals.*

**Response:** Thank you for your comment. The SP2 signal is recorded based on the elapsed time, with each time window corresponding to information about a single particle. For each particle, the corresponding original scattering signal and incandescence signal are both 100-dimensional. The position of particles within the instrument is not known in advance. Among SP2's four detectors, there is a two-element APD (TEAPD) detector. This detector has a gap perpendicular to the particle's direction of motion, resulting in a notch in the TEAPD signal, as shown in Fig. R1a. Given the stability of SP2's optical alignment and constant sample flow rate, this notch provides a precise time reference for a particle's position within the instrument. In practice, the signal from the leading element is inverted, transforming the notch into a zero-crossing point (Fig. R1b) (Gao et al., 2007). Since SP2 simultaneously records data from all four detector channels, this time reference is valid for the signals from the other three detectors as well.

[Figure]

**Figure R1.** (a) The original scattering signal measured by TEAPD before the signal from the leading element is inverted. (b) The TEAPD signal obtained by SP2, with the blue asterisk indicating the position of the zero-crossing point.

For pure scattering particles, we locate the zero-crossing point in the scattering signal and then extract 22 data points both before and after it, creating a 45-dimensional feature dataset (Fig. R2a). In this newly constructed 45-dimensional feature signal, particles are positioned at consistent locations within the instrument for each corresponding dimension, eliminating the influence of laser intensity distribution on the scattering signal.

When inverting the $D_c$ of BC-containing particles, since the peak intensity of the incandescence signal is positively correlated with the mass of the refractory BC

component in the particle, the peak of the incandescence signal is selected as a reference point, from which 22 data points are extracted both preceding and following this point (Fig. R2b), yielding a 45-dimensional feature dataset. This method ensures the incandescence signal peaks from different BC-containing particles are positioned at the same dimension within the feature dataset, facilitating direct comparisons between particles. The optical properties of externally mixed BC are also determined by the refractory BC component, so the same feature selection method is adopted.

For internally mixed BC, its size and optical cross-section characteristics are reflected by both the scattering and incandescence signals. We select scattering features for internally mixed BC using the same approach as for pure scattering particles. Considering that the relative relationship between the incandescence and scattering signals in the original signal can reflect particle characteristics, the incandescence signal is selected with 22 data points before and after the zero-crossing point in a similar way (Fig. R2c). The 90-dimensional feature signal composed of the selected 45-dimensional scattering signal and 45-dimensional incandescence signal is used as the input feature for the internally mixed BC inversion model.

[Figure]

**Figure R2.** Relationship between the original SP2 signals (line plots) and the feature signals used in machine learning model construction (scatter plots) for different particle types: (a) pure scattering particles; (b) externally mixed BC; (c) internally mixed BC. The method for selecting feature signals used in inverting the core diameter ($D_c$) of BC-containing particles is identical to that used for externally

mixed BC.

We have provided additional descriptions of the principles of signal feature selection for different particle types as suggested. The Figs. R1 and R2 presented here have also been incorporated into the revised manuscript and Supplement Information to further illustrate these principles. Specifically, Fig. R1 has been added as Fig. 3 in the revised manuscript, while Fig. R2 has been included as Fig. S1 in the Supplement Information.

Lines 141 to 143:

"*The SP2 signal is recorded based on the elapsed time, with each time window corresponding to information about a single particle. For each particle, the original scattering signal and incandescence signal are both 100-dimensional. The position of particles within the instrument is not known in advance.*"

Lines 154 to 162:

"*As mentioned in Sect. 2.2, one of the four detectors in the SP2 is a split APD detector. This detector has a gap perpendicular to the particle's direction of motion, resulting in a notch in the TEAPD signal, as shown in Fig. 3a. Given the stability of SP2's optical alignment and constant sample flow rate, this notch provides a precise time reference for a particle's position within the instrument. In practice, the signal from leading element is inverted, transforming the notch into a zero-crossing point (Fig. 3b) (Gao et al., 2007).* Since SP2 simultaneously records data from all four detector channels, this time reference is valid for the signals from the other three detectors as well. *We locate the zero-crossing point in the scattering signal and then extract 22 data points both before and after it, creating a 45-dimensional feature dataset (Fig. S1a). Through this standardization, the differences in signal intensity can be accurately attributed to the inherent physical properties of the particles.*

Lines 168 to 173:

*Based on this characteristic, the peak of the incandescence signal is selected as a reference point, from which 22 data points are extracted both preceding and following this point (Fig. S1b), yielding a 45-dimensional feature dataset used for inverting the $D_c$ of BC-containing particles. This method ensures the incandescence signal peaks from different BC-containing particles are positioned at the same dimension within the feature dataset, facilitating direct comparisons between particles while preserving comprehensive information about the incandescence process.*

Lines 184 to 189:

"*Compared with other particle types, the internally mixed BC has a more complex structure. ... Simultaneously, considering that the relative relationship between the original incandescence and scattering signals can reflect particle characteristics, such as coating thickness (Moteki and Kondo, 2007; Schwarz et al., 2006; Subramanian et al., 2010), the incandescence signal is selected with 22 data points before and after the zero-crossing point in a similar way (Fig. S1c). The feature extraction process yields a 90-dimensional feature dataset, comprising 45-dimensional scattering signal and 45-dimensional incandescence signal, ensuring that we can*

*comprehensively capture the key characteristics of internally mixed BC."*

*4) Line 191: What is "GridSearchCV function".*

**Response:** Thank you for your question. GridSearchCV is an automated hyperparameter tuning technique that combines grid search with cross-validation (Ahmad et al., 2022). In this process, a range of values is defined for each hyperparameter, and then all possible combinations are systematically tried. It uses K-fold cross-validation to evaluate the performance of each hyperparameter combination, eliminating the need for manual adjustments by automatically executing tests for all possible combinations. GridSearchCV assesses performance based on the average results of cross-validation and ultimately selects the hyperparameter combination that produces the best performance. The main advantage of this method lies in its ability to comprehensively and systematically explore the hyperparameter space while providing reliable estimates of the model's generalization capability. We have added a brief explanation of the GridSearchCV function in the revised manuscript.

Lines 251 to 254:

*"The GridSearchCV with 5-fold cross-validation is employed to optimize the hyperparameter configuration of the LightGBM inversion model.* ***This comprehensive approach facilitates an exhaustive search for the optimal hyperparameter combination within a predefined parameter space (Ahmad et al., 2022). By utilizing cross-validation, the methodology effectively mitigates the risk of overfitting and provides robust estimates of the model's generalization performance."***

*5) Line 191: Please check the grammar of the sentence starting with "Based on". What is the subject of this sentence?*

**Response:** Thank you for your careful review and for pointing out this grammatical issue. The sentence beginning with "Based on" lacks a clear subject, which affects its grammatical structure. We have revised the sentence to "The GridSearchCV with 5-fold cross-validation is employed to optimize the hyperparameter configuration of the LightGBM inversion model." [Page 10, Line 251]

**Response:** Thank you for your comment. We have expanded the rationale for choosing the LightGBM algorithm in the introduction section of the revised manuscript. Regarding the abstract, we have made a brief supplement to address this point as well.

The relevant amendments of the introduction are detailed on Lines 55 to 66 :

*"As an alternative, data-driven models such as machine learning (ML) can provide a good supplement to physical process-based models. ML can efficiently capture the nonlinear relationship between inputs and outputs, and has found widespread application in various fields (Carleo et al., 2019; Jordan and Mitchell, 2015; Liakos et al., 2018; Tarca et al., 2007). **In recent years, tree-based machine learning models have gained considerable popularity due to their extremely high computational speed, satisfactory accuracy, and interpretability (Keller and Evans, 2019; Li et al., 2022; Wei et al., 2021; Yang et al., 2020). Among these, the Light Gradient Boosting Machine (LightGBM) has shown particularly outstanding performance. As a novel distributed gradient boosting framework based on decision tree algorithms, LightGBM can extract information from data more effectively than traditional tree models, excelling in handling complex non-linear relationships and high-dimensional features (Ke et al., 2017; Liu et al., 2024; Zhong et al., 2021).***

*It employs innovative techniques such as gradient-based one-side sampling (GOSS) and exclusive feature bundling (EFB), which significantly improve computational efficiency while maintaining high predictive performance (Ke et al., 2017; Sun et al., 2020). Furthermore, different from some black-box models, LightGBM maintains the interpretability characteristic of tree-based models (Gan et al., 2021; Zhang et al., 2019), which can provide decision path analysis, allowing for deeper insights into the decision-making process. Considering these advantages, LightGBM can be an ideal tool for analyzing large SP2 datasets and inverting BC mixing states."*

Lines 20 in abstract:

*"However, the derivation of BC mixing state from SP2 is quite challenging. Since the SP2 records individual particle signals, it requires complex data processing to convert raw signals into particle size and mixing states. Besides, the rapid accumulation of substantial data volumes impedes real-time analysis of BC mixing states. This study employs a light gradient boosting machine (LightGBM), an advanced tree-based ensemble learning algorithm, to establish an inversion model which directly correlates SP2 signals with the mixing state of BC-containing particles."*

*2) Expand the discussion on how this method improves upon or complements existing techniques, such as the LEO fitting method or other machine learning approaches. For example, what specific challenges of previous methods (e.g., noise resistance, scalability) does this model overcome?*

**Response:** Thank you for your suggestion. A comparison between the LEO fitting results and machine learning results is shown in Fig. 9. As we discussed in the manuscript, the machine learning method utilizes more complete SP2 signals, enabling a more comprehensive characterization of particles. Furthermore, the signals used in the machine learning method have a higher signal-to-noise ratio, making it more robust against background noise compared to the LEO fitting method. The detailed discussion can be found on Lines 393 to 400 of the revised manuscript and attached below:

*"Figure 9 illustrates the LEO fitting results for two different BC-containing particles. Despite nearly identical leading-edge data resulting in similar Gaussian distributions and consequently the same $D_p$ values through LEO fitting, the complete scattering signals of these particles exhibit significant differences. The ML model, by incorporating these distinctive signal features, can effectively capture these variations, leading to different $D_p$ predictions. Moreover, the leading edge is traditionally defined as the signal from baseline-subtracted zero up to 5 % of the maximum laser intensity (Taylor et al., 2015). As shown in Fig. 9, this portion of the signal (in the grey-shaded area) is close to the baseline, making it particularly susceptible to noise interference. Compared to LEO fitting method, the ML model utilized a broad range of signals with a high signal-to-noise ratio, demonstrating enhanced noise resistance."*

[Figure]

*"**Figure 9.** Comparison of the scattering signal used in the $D_p$ inversion process for internally mixed BC and corresponding calculation results from both the LEO fitting and the ML methods. The solid line represents the scattering signal obtained by SP2, and the part marked with solid dots is the scattering signal input to the ML model. The gray shaded area shows the leading-edge data used in the LEO fitting process, and the dashed line represents the scattering signal of the original particle reconstructed by LEO fitting."*

*3) Although the study uses data from a single site, could the authors discuss the expected performance of the model in different environmental conditions or geographical regions? For example, how might differences in aerosol composition affect results?*

**Response:** Thank you for your suggestion. Operationally, SP2 data calibration is essential when conducting measurements in different regions due to variations in instrument status. Physical inversion methods inherently require these calibration procedures to ensure accuracy across different observational settings. Similarly, machine learning algorithms, including our approach, also necessitate comparable calibration processes. For instance, increased voltage leads to stronger laser intensity, resulting in enhanced SP2 scattering signals, which can cause the model to overestimate particle sizes. To address this, a voltage-related calibration function can be introduced to adjust the model's predictions. By integrating these calibrations derived from experimental data, we can enhance the model's robustness and ensure its applicability across a wide range of observational settings.

*4) Consider adding uncertainty quantification for model predictions. For instance, providing confidence intervals for particle size or optical property predictions would help assess reliability in practical applications.*

**Response:** Thank you for your comment. We have added an analysis of the prediction

errors of the BC mixing state inversion model across different particle sizes. By examining the distribution of prediction errors for the $D_p$ of internally mixed BC in conjunction with the number size distribution (Fig. 5), we demonstrate that the model developed in this study achieves high accuracy in the 150–300 nm size range, where particles are most concentrated. Furthermore, based on the 25% and 75% percentiles of the error distribution, the model's prediction errors exhibit minimal fluctuation within this size range. We have also explained the increased prediction errors at both ends of the size distribution. This analysis helps to illustrate the model's performance across different particle size ranges and highlights its strengths and limitations. The added analysis can be found on Lines 320 to 333 in the revised manuscript and attached below:

*"To comprehensively assess the model's performance across different particle size ranges, we further analyzed the prediction error distribution for $D_p$ inversion model of internally mixed BC, as shown in Fig. 5. For particles smaller than 150 nm, the prediction errors average around 4 nm, primarily due to the low signal-to-noise ratio of their scattering signals, which introduces larger uncertainties in the LEO fitting process. The model exhibits optimal performance for particles between 150 nm and 300 nm, with an average prediction error of approximately 1.5 nm. Furthermore, based on the 25% and 75% percentiles of the error distribution, the model's prediction errors exhibit minimal fluctuation within this size range. However, prediction errors gradually increase with particle size, becoming particularly significant for particles larger than 480 nm. This trend can be attributed to occasional irregular signals at larger sizes, such as scattering or incandescence signals with abnormally broad peak widths. These signal irregularities pose challenges to the accurate characterization of particle physical properties, affecting both LEO fitting accuracy and ML model predictions, potentially leading to more pronounced discrepancies between the two methods. The number size distribution of internally mixed BC in the testing set indicates that most particles fall within the 150–300 nm range, where the model demonstrates highest accuracy. Although the prediction errors are relatively larger at both ends of the size distribution (< 150 nm and > 400 nm), the number of particles in these ranges is comparatively small, thus having limited impact on the overall performance of the model."*

[Figure]

*"Figure 5. The prediction error distribution for $D_p$ inversion model of internally mixed BC, and normalized number size distribution for $D_p$ of internally mixed BC in the testing set. The solid lines in error distribution represent the median value, the upper and lower boundary of the shaded area is between the 25% and 75% quantiles."*

*5) The SHAP analysis identifies important features in the scattering and incandescence signals. However, the physical relevance of these features (e.g., why certain regions of the signal are more predictive) could be discussed more thoroughly.*

**Response:** Thank you for pointing this out. Through SHAP analysis, it can be observed that several crucial scattering signal features are distributed near the peak. This part of the signal represents a non-linear combination of coating evaporation and incident laser intensity changes. The pronounced signal variability within this certain region enables the machine learning model to discriminate and extract distinctive features across different particles during the training process.

Regarding the important incandescence signal features, they are primarily concentrated in the interval spanning from the initial rise of the incandescence signal to its peak intensity. The changes in the incandescence signal are closely related to the refractory BC component in BC-containing particles, thereby providing insights into the comprehensive characteristics of the entire particle.

We have further elaborated on the physical significance of the important features indicated by SHAP analysis in the revised manuscript. Specific modifications can be found on lines 367 to 371 of the revised manuscript:

[revised manuscript text omitted]

***Response to the comments in the quick report of Reviewer #2 (EGUSPHERE-2024-2496)***

*The manuscript "Inversion Algorithm of Black Carbon Mixing State Based on Machine Learning" presents a novel approach for deriving the mixing state of black carbon (BC) particles using a machine learning (ML) method, specifically LightGBM, as an alternative to traditional physical inversion methods. The work is significant as it addresses the challenge of processing large volumes of SP2 data for real-time analysis. The integration of SHapley Additive exPlanation (SHAP) for feature importance analysis adds a valuable layer of interpretability to the model. This paper presents a valuable contribution to the field of atmospheric science, particularly in improving the efficiency of BC mixing state analysis through machine learning. With improvements in the methodology description, error analysis, and discussion of model limitations, this paper could make a stronger impact in its field. However, there are several areas where the manuscript could be improved for clarity, reproducibility, and scientific rigor. The problems are addressed as following.*

**Response:** We sincerely thank the reviewer's comprehensive and helpful comments. In response, we have refined the manuscript by incorporating more precise definitions of different particle types, expanding the methodological description in Section 3 "Machine-learning-based inversion algorithm", and enhancing the analysis in the result section. Please find our point-by-point responses listed below. The reviewer's comments are in *Italic* followed by our responses and revisions (in blue).

*1) The introduction provides sufficient background on the importance of BC mixing states and the challenges of deriving such information from SP2 data. However, the transition to the machine learning method feels abrupt. The motivation for choosing LightGBM over other machine learning models should be better justified. For instance, why was LightGBM preferred over other common models like Random Forest or Neural Networks?*

**Response:** This comment aligns with Comment 1 in the main comments of Reviewer #2. We appreciate the reviewer's suggestion and have accordingly expanded the introduction to provide more rationale for our choice of LightGBM. Specific modifications can be found on Lines 55 to 66 of the revised manuscript, please refer to our response to Comment 1 in the main comments of Reviewer #2.

*2) Some progresses are made about the complex mixing structures and morphologies (Wang et al., 2021b; Wang et al., 2017; Wang et al., 2021a; Pang et al., 2022). This should also be mentioned when considering the accuracy of LEO and ML inversion*

**Response:** Thanks for your suggestion. The complex structure and morphology of BC-

containing particles indeed affect the accuracy of the inversion to some extent. We have added this information to the result analysis section.

Lines 326 to 329:

*"__Furthermore, the complex mixing structures and morphologies of BC-containing particles also affect the accuracy of both methods (Pang et al., 2022; Wang et al., 2017, 2021a, b). Given that the quantitative impact of these factors is challenging to determine currently, this study does not involve discussion related to this aspect, leaving it as a direction for future research.__"*

*3) The description of feature extraction from SP2 signals is somewhat unclear. It is mentioned that 45-dimensional data are used, but the criteria for selecting these dimensions and their physical significance should be explained more thoroughly.*

**Response:** Thank you for pointing this out. The SP2 signal is recorded based on the elapsed time, with each time window corresponding to information about a single particle. For each particle, the corresponding original scattering signal and incandescence signal are both 100-dimensional. The position of particles within the instrument is not known in advance. Among SP2's four detectors, there is a two-element APD (TEAPD) detector. This detector has a gap perpendicular to the particle's direction of motion, resulting in a notch in the TEAPD signal, as shown in Fig. R1a. Given the stability of SP2's optical alignment and constant sample flow rate, this notch provides a precise time reference for a particle's position within the instrument. In practice, the signal from the leading element is inverted, transforming the notch into a zero-crossing point (Fig. R1b) (Gao et al., 2007). Since SP2 simultaneously records data from all four detector channels, this time reference is valid for the signals from the other three detectors as well.

[Figure]

**Figure R3.** (a) The original scattering signal measured by TEAPD before the signal from the leading element is inverted. (b) The TEAPD signal obtained by SP2, with the blue asterisk indicating the position of the zero-crossing point.

For pure scattering particles, we locate the zero-crossing point in the scattering signal and then extract 22 data points both before and after it, creating a 45-dimensional feature dataset (Fig. R2a). In this newly constructed 45-dimensional feature signal, particles are positioned at consistent locations within the instrument for each corresponding dimension, eliminating the influence of laser intensity distribution on the scattering signal.

When inverting the BC core diameter ($D_c$) of BC-containing particles, since the peak intensity of the incandescence signal is positively correlated with the mass of the refractory BC component in the particle, the peak of the incandescence signal is selected as a reference point, from which 22 data points are extracted both preceding and following this point (Fig. R2b), yielding a 45-dimensional feature dataset. This method ensures the incandescence signal peaks from different BC-containing particles are positioned at the same dimension within the feature dataset, facilitating direct comparisons between particles. The optical properties of externally mixed BC are also determined by the refractory BC component, so the same feature selection method is adopted.

For internally mixed BC, its size and optical cross-section characteristics are reflected by both the scattering and incandescence signals. We select scattering features for internally mixed BC using the same approach as for pure scattering particles. Considering that the relative relationship between the incandescence and scattering signals in the original signal can reflect particle characteristics, the incandescence signal is selected with 22 data points before and after the zero-crossing point in a similar way (Fig. R2c). The 90-dimensional feature signal composed of the selected 45-dimensional scattering signal and 45-dimensional incandescence signal is used as the input feature for the internally mixed BC inversion model.

[Figure]

**Figure R4.** Relationship between the original SP2 signals (line plots) and the feature signals used in

machine learning model construction (scatter plots) for different particle types: (a) pure scattering particles; (b) externally mixed BC; (c) internally mixed BC. The method for selecting feature signals used in inverting the core diameter ($D_c$) of BC-containing particles is identical to that used for externally mixed BC.

We have provided additional descriptions of the principles of signal feature selection for different particle types as suggested. The Figs. R1 and R2 presented here have also been incorporated into the revised manuscript and Supplement Information to further illustrate these principles. Specifically, Fig. R1 has been added as Fig. 3 in the revised manuscript, while Fig. R2 has been included as Fig. S1 in the Supplement Information.

Lines 141 to 143:

*"**The SP2 signal is recorded based on the elapsed time, with each time window corresponding to information about a single particle. For each particle, the original scattering signal and incandescence signal are both 100-dimensional. The position of particles within the instrument is not known in advance.**"*

Lines 154 to 162:

*"**As mentioned in Sect. 2.2, one of the four detectors in the SP2 is a split APD detector. This detector has a gap perpendicular to the particle's direction of motion, resulting in a notch in the TEAPD signal, as shown in Fig. 3a. Given the stability of SP2's optical alignment and constant sample flow rate, this notch provides a precise time reference for a particle's position within the instrument. In practice, the signal from leading element is inverted, transforming the notch into a zero-crossing point (Fig. 3b) (Gao et al., 2007).** Since SP2 simultaneously records data from all four detector channels, this time reference is valid for the signals from the other three detectors as well. **We locate the zero-crossing point in the scattering signal and then extract 22 data points both before and after it, creating a 45-dimensional feature dataset (Fig. S1a). Through this standardization, the differences in signal intensity can be accurately attributed to the inherent physical properties of the particles.**

Lines 168 to 173:

*__Based on this characteristic, the peak of the incandescence signal is selected as a reference point, from which 22 data points are extracted both preceding and following this point (Fig. S1b), yielding a 45-dimensional feature dataset used for inverting the $D_c$ of BC-containing particles. This method ensures the incandescence signal peaks from different BC-containing particles are positioned at the same dimension within the feature dataset, facilitating direct comparisons between particles while preserving comprehensive information about the incandescence process.__*

Lines 184 to 189:

*"Compared with other particle types, the internally mixed BC has a more complex structure. ... **Simultaneously, considering that the relative relationship between the original incandescence and scattering signals can reflect particle characteristics, such as coating thickness (Moteki and Kondo, 2007a; Schwarz et al., 2006; Subramanian et al., 2010), the incandescence signal is**

*selected with 22 data points before and after the zero-crossing point in a similar way (Fig. S1c). The feature extraction process yields a 90-dimensional feature dataset, comprising 45-dimensional scattering signal and 45-dimensional incandescence signal, ensuring that we can comprehensively capture the key characteristics of internally mixed BC.”*

*4) The split of the dataset (70/30) for training and testing is standard, but additional information on how the dataset was balanced or handled for bias should be included. Provide more detail on the characteristics of the training and testing sets (e.g., balance of particle types, particle sizes). If any resampling techniques (such as SMOTE) were used to handle imbalances, this should be mentioned.*

**Response:** Thank you for your reminder. In this study, the dataset used for machine learning comprises 15 days of SP2 field observation data. For each particle type, the number of samples used in machine learning reaches an order of $10^5$. The dataset is randomly partitioned into training and testing sets with a ratio of 7:3, and this unbiased selection method helps improve the reliability and generalizability of the model. To demonstrate the effectiveness of this data division, we analyzed the normalized number size distributions of particle diameter ($D_p$) in both the training and testing sets for the internally mixed BC inversion model. As shown in Fig. R3, the consistent distributions between these two sets validate the rationality of our data partitioning approach. The related description and Fig. R3 have been included as Fig. S2 in the Supplement Information.

[Figure]

**Figure R3.** The normalized number size distribution of the training set (black marks and line) and testing set (red marks and line) used in the $D_p$ inversion model for internally mixed BC.

*5) While the high R² values are impressive, the paper lacks a more detailed error analysis. The discussion primarily focuses on RMSE and MAE, but additional insights on how these errors are distributed across different particle sizes or conditions would enhance the results section. Consider providing a deeper analysis of the error distribution, perhaps by including more examples of model underperformance and an explanation for these cases.*

**Response:** This comment closely aligns with the suggestions in Comment 4 of Reviewer #2's main comments regarding a deeper analysis of model predictions. In

response, we have added a more comprehensive prediction error analysis for the $D_p$ inversion model of internally mixed BC, as it is the most representative aspect of our study. This detailed analysis can be found on Lines 320 to 333 of the revised manuscript. For a full discussion of this analysis, please refer to our response to Comment 4 in Reviewer #2's main comments.

*6) The traditional processing methods of SP2 detection signals should be discussed in detail, such as how externally mixed black carbon is identified, how internally mixed black carbon is determined, and the specific methods and principles for calculating $D_p$ and $D_c$. This would help readers better understand the data in the results and discussion sections.*

**Response:** Thank you for your suggestion. Operationally, the classification between externally and internally mixed BC is determined by the time delay, defined as the time difference between the peak of the incandescence signal and the scattering signal. In the revised manuscript, we have provided a more detailed explanation of how the traditional SP2 signal processing method distinguishes between externally mixed BC and internally mixed BC.

The relevant amendments are detailed on Lines 119 to 122:

"***The incandescence signal peak occurs when all non-BC material has evaporated and the BC reaches its incandescence temperature, thus the magnitude of $\Delta t$ correlates with the thickness of the coating on BC particles: a larger $\Delta t$ corresponds to a thicker coating that takes longer to evaporate.*** *By examining the distribution of $\Delta t$ values in the SP2 measurements, as illustrated in Fig. 2d (Sedlacek et al., 2012; Subramanian et al., 2010; Zhang et al., 2016), BC-containing particles with $\Delta t < 2 \mu s$ are classified as externally mixed BC (Fig. 2b), while those with $\Delta t \geq 2 \mu s$ are categorized as internally mixed BC (Fig. 2c)."*

In addition, in the "Construction of label dataset" section, we have provided a more detailed introduction to the traditional processing methods and principles for inverting the particle size of different particle types based on SP2 detection signals.

Lines 196 to 198:

"***To obtain the $D_p$ of pure scattering particles, the scattering signal amplitude is first used to determine the particle's scattering cross-section, which is then compared with that of PSL particles of known diameter to determine the $D_p$.***"

Lines 202 to 205:

*"The peak intensity of thermal radiation emitted by the rBC is proportional to its mass ($M_{BC}$) (Moteki and Kondo, 2007b). According to the empirical relationship between the incandescent light intensity and the particle mass calibrated using fullerene soot, the $M_{BC}$ of each BC-containing particle can be quantified. Assuming a density of 1.8 g cm$^{-3}$ (Bond and Bergstrom, 2006), the measured $M_{BC}$ can be further converted into the mass-equivalent diameter $D_c$."*

Lines 208 to 215:

*"As the evaporation of the particle, the scattering signal deviates from a Gaussian distribution, making it inappropriate to directly use the scattering amplitude to calculate $D_p$.* **_To properly size these particles, the LEO fitting method is employed to reconstruct the Gaussian signal. As described in Sect. 3.3, the zero-crossing point in the TEAPD signal can serve as a position reference for particles in the SP2. Moreover, the position difference between the zero-crossing point and the peak laser intensity remains constant during measurements. The width of the laser intensity distribution and the position of peak laser intensity relative to the zero-crossing point, both determined by Gaussian fitting of numerous unsaturated pure scattering particles, are used to constrain the LEO fitting, leaving the fitting amplitude as the only free parameter. Using leading-edge data from the signal onset to 5% of the maximum laser intensity for LEO fitting, can obtain the reconstructed scattering amplitude and further convert it to particle scattering cross-section._** *The $D_p$ of internally mixed BC can be derived by inputting the LEO-fitted scattering cross-section, BC core diameter, and the corresponding refractive indices of the core and coating into the Mie calculation model (Laborde et al., 2012; Liu et al., 2014; Schwarz et al., 2008; Taylor et al., 2015)."*

*7) Is it possible that the authors provide signal classification and discrimination in the machine learning process, including the determination of the zero-crossing point in the TEAPD signal, and then match the corresponding machine learning scheme? This could eliminate the need for manual preprocessing, making the method more user-friendly.*

**Response:** Thank you for your suggestion. As mentioned in the "Machine-learning-based inversion algorithm" section of the manuscript, pure scattering particles and BC-containing particles are distinguished based on a set threshold for the incandescence signal peak amplitude. BC-containing particles are further classified into externally and internally mixed BC based on the time delay. The zero-crossing point is obtained through interpolation of the TEAPD signal. These processes have been developed into a robust algorithm, which can be executed by simply inputting the raw SP2 signals and completed in a short time. In the future, we plan to integrate this part with machine learning methods to enhance the overall system's user-friendliness.

*8) In Figure 3, there are three types of particles: pure scattering particles, BC containing particles, and internally mixed BC; In Figure 4, there are pure scattering particles, externally mixed particles, and internally mixed BC. Then, did BC containing particles include both internally mixed and surface-attached BC particles? Or did internally mixed BC particles excluded surface-attached BC particles? The definition should be clear when they were first mentioned.*

**Response:** Thank you for pointing this out. In this study, ambient particles measured by SP2 are classified into pure scattering particles and BC-containing particles. Pure scattering particles are those that only scatter light without significant absorption, while BC-containing particles, which contain refractory BC (rBC), both scatter and absorb

light. BC-containing particles are further subdivided into externally mixed BC and internally mixed BC. Externally mixed BC refers to freshly emitted BC particles that have not yet mixed with other aerosol components, while internally mixed BC describes BC that has undergone atmospheric aging processes and incorporated with other materials (Oshima et al., 2009). The conceptual definitions of these particle types have been incorporated into the revised manuscript, specifically elaborated on Lines 110 to 115.

Operationally, we differentiate the pure scattering particle and BC-containing particle depending on whether it has the incandescence signal. BC-containing particles are further classified into externally and internally mixed BC based on the time delay.. A more comprehensive explanation of these classification criteria is provided in our response to Comment 6.

According to the aforementioned definitions for different particle types, the "surface-attached" BC type you referred to belongs to the internally mixed BC. However, current analyses of SP2 signals, such as time delay, remain insufficient to conclusively determine particle morphology (Sedlacek et al., 2015). Therefore, we did not make a distinction for the "surface-attached" type in this study.

We have provided additional clarification on this point in the revised manuscript, specifically on Lines 125 to 127:

"*Additionally, relying solely on time delay may not be sufficient to distinguish certain types of BC-containing particles, such as "attached type" (Sedlacek et al., 2015). Therefore, in this study, no further classification is made regarding the detailed morphology of BC-containing particles.*"

*9) Lines 168-169: Why your chose 170 nm as the lower limit for $D_p$? Please present the references.*

**Response:** Thank you for your question. This is determined by the detection limit of the SP2. For pure scattering particles, when the $D_p$ is small, the corresponding scattering cross-section is insufficient to produce a strong enough scattering signal. These weak signals are easily affected by the background noise, and as a result, the particle size inverted may not be accurate. During preprocessing, we examined the original scattering signals corresponding to pure scattering particles of different sizes. We ultimately selected 170 nm as the lower limit for the $D_p$ to ensure the quality of data used in the machine learning process (Schwarz et al., 2006; Sedlacek et al., 2015).

We have added the references on Line 228 in the revised manuscript:

"*For the pure scattering particles, the smallest size limit for $D_p$ is set at 170 nm (Schwarz et al., 2006; Sedlacek et al., 2015).*"

*10) Figure 3: It seems the particle size inversion of internally mixed BC particles can have deviations up to 100 nm. Maybe this deviation is too high, I think. Why this*

*happens?*

**Response:** Thank you for your comment. This observed phenomenon is due to occasional particles with irregular signals at the large size end, characterized by scattering or incandescence signals with peak widths significantly broader than typical measurements. In these cases, the physical characteristics of the particles are difficult to accurately characterize, which consequently impacts the accuracy of both LEO fitting and ML model predictions, potentially introducing more pronounced discrepancies between the two methods.

We have incorporated a detailed explanation in the error analysis for the $D_p$ inversion of internally mixed BC on Lines 315 to 320 in the revised manuscript.

*"However, prediction errors gradually increase with particle size, becoming particularly significant for particles larger than 480 nm. This trend can be attributed to occasional irregular signals at larger sizes, such as scattering or incandescence signals with abnormally broad peak widths. These signal irregularities pose challenges to the accurate characterization of particle physical properties, affecting both LEO fitting accuracy and ML model predictions, potentially leading to more pronounced discrepancies between the two methods."*

*11) Figure3: Why the pure scattering and internally mixed ones are smaller and none of them are ~600 nm. Is that true? The upper limit of the $D_c$ of BC-containing particles is up to 600 nm, then why the upper limit of the $D_p$ of internally mixed BC is smaller than 600 nm?*

**Response:** Thank you for your question. The SP2 can detect pure scattering particles and internally mixed BC with diameters up to 600 nm. However, for relatively large particles, the scattered light may exceed the threshold of the scattering signal detector, as illustrated in Fig. R4. This figure shows that the scattering signal information for such particles is incomplete. While the LEO fitting method can reconstruct the original scattering signal, inputting these incomplete signals into machine learning models may compromise model performance. Consequently, in this study, particle data with saturated scattering signals are not included in the machine learning model. This is discussed in the "Construction of label dataset" section.

For BC-containing particles, the BC core diameter ($D_c$) is calculated using the peak intensity of the incandescence signal. This calculation method is relatively straightforward, and typically, the incandescence signal does not reach saturation when $D_c$ is around 600 nm. As a result, in this study, the $D_c$ of BC-containing particles can reach up to 600 nm.

Previous SP2 studies have predominantly focused on pure scattering particles within the 200–400 nm range, with $D_p$ of internally mixed BC predominantly distributed below 500 nm (Huang et al., 2012; Liu et al., 2014; Zhang et al., 2018). The particle samples selected for this study cover these ranges.

[Figure]

**Figure R4.** The scattering and incandescence signals of the saturated particle.

*12) Figure 3c: Was the larger deviation due to the complex mixing structures and morphologies? When people use core-shell model to calculate the size, they omitted the differences of mixing structures and morphologies among individual BC particles. So, I think the deviation came from the simplification of the shape model of BC.*

**Response:** Thank you for your comment. BC-containing particles have complex structures and morphologies, while traditional physical inversion is based on simplified core-shell model assumptions. This simplification may lead to discrepancies between machine learning predictions and physical inversion results. However, under current field observation conditions, it is challenging to accurately quantify the impact of BC structure and morphology on the results. This issue requires more in-depth investigation in future studies to better understand and assess the influence of these complex factors on inversion results. We have added the relevant explanation on Lines 336 to 339 in the revised manuscript.

*13) How about the SHAP summary for the $D_p$ and $D_c$ externally mixed BC particles and the SHAP for $C_{sca}$ and $C_{abs}$ for different types of BC particles?*

**Response:** Thank you for your question. According to the definition of externally mixed BC, the $D_c$ and $D_p$ of externally mixed BC are identical. As a result, the size inversion for externally mixed BC is equivalent to determining the $D_c$ of BC-containing particles. Given this equivalence, for externally mixed BC, this research focuses only on the inversion of its optical cross-section.

For pure scattering particles, the $D_p$ can be directly retrieved from the scattering amplitude. Consequently, in the SHAP analysis results, the feature dimensions corresponding to the scattering signal peak are of key importace. Similarly, since the refractory BC mass is proportional to the peak value of the incandescence signal, there is a strong correlation between the $D_c$ and the amplitude of the incandescence signal. This relationship is evident in the SHAP results, where feature dimensions associated with the incandescence signal peak demonstrate significant importance. For externally mixed BC, the optical cross-sections are calculated using Mie theory based on $D_c$.

Therefore, the SHAP analysis results also show that the feature dimensions corresponding to the peak position of the incandescence signal are crucial.

The case of internally mixed BC is more complex. Its optical cross-section exhibits a complex, non-linear relationship with both scattering and incandescence signals. Figures R5a and R5b illustrate the SHAP summary plots for the inversion models of absorption and scattering cross-sections of internally mixed BC, respectively. Due to the small magnitude of the optical cross-section, the SHAP values shown here have been amplified for clarity. The SHAP summary plot for the absorption cross-section of internally mixed BC shows that most of the top fifteen important features are related to the incandescence signal. Specifically, features BBLG28, BBLG29, and BBLG27, corresponding to the feature dimensions near the peak of the incandescence signal, contribute most significantly to the absorption cross-section. Among the top fifteen features, several features related to the scattering signal (SCLG12 to SCLG15) are also present. These features, associated with the peak positions of the scattering signal, demonstrate that larger feature values positively contribute to the inversion of the absorption cross-section. This phenomenon can be attributed to the "lensing effect" of the coating, which enhances the absorption of the BC core (Cappa et al., 2012; Schwarz et al., 2008). For the inversion model of the scattering cross-section of internally mixed BC, scattering signal features are particularly important, showing a positive correlation with the scattering cross-section. Meanwhile, the incandescence signal, which reflects the characteristics of the BC core, also plays an important role in the inversion process.

[Figure]

**Figure R5.** The SHAP summary plot for the optical cross-section inversion model of internally mixed BC: (a) absorption cross-section; (b) scattering cross-section.

We have supplemented the SHAP results for the inversion models of the optical cross-section of internally mixed BC in the Supporting Information. While inversion models for particle size and optical properties of other particle types typically exhibit relatively

straightforward relationships between physical properties and signal features. This study primarily focuses on the more complex case of internally mixed BC, and does not elaborate on the SHAP results of these relatively simple scenarios.

*14) Figure 8: Is the axis of this graph logarithmic?*

**Response:** Thank you for your careful review. Indeed, Fig. 8 utilizes a logarithmic scale for its axes. To clarify, we have added a clear note in the figure caption specifying that the axes are plotted on a logarithmic scale:

*"**Figure 10.** Distribution of $D_p$ and $D_c$ of internally mixed BC retrieved from the BC mixing state inversion model. The main panel is a two-dimensional histogram where the color represents the normalized number of particles within a specific size range. Side panels display the normalized number size distributions of $D_c$ and $D_p$, each scaled to its peak value. **Both $D_p$ and $D_c$ axes use logarithmic scales.**"*

*15) Figure 9: I think the $D_p/D_c$ and $D_c$ values can also be provided. Because these values are import for the analyzing of BC aging.*

**Response:** Thank you for your suggestion. We have added the $D_p/D_c$ and $D_c$ values in Fig. 9. The revised figure is shown as follows:

[revised manuscript text omitted]

Response to Particle Mixing Structure, Geophysical Research Letters, 48, e2021GL096437, https://doi.org/10.1029/2021GL096437, 2021b.

Zhang, Y., Su, H., Ma, N., Li, G., Kecorius, S., Wang, Z., Hu, M., Zhu, T., He, K., Wiedensohler, A., Zhang, Q., and Cheng, Y.: Sizing of Ambient Particles From a Single-Particle Soot Photometer Measurement to Retrieve Mixing State of Black Carbon at a Regional Site of the North China Plain, JGR Atmospheres, 123, https://doi.org/10.1029/2018JD028810, 2018.

*Response to the comments of Xiaolong Fan (EGUSPHERE-2024-2496)*

*The Single-particle soot photometer (SP2) is a widely recognized instrument for quantifying the mixing state of black carbon (BC). However, deriving BC mixing state from SP2 measurements remains challenging. This study introduces a user-friendly SP2 inversion method based on machine learning. Notably, the machine learning approach does not merely replicate the results of physical inversion methods but also utilizes previously unexploited signals. It overcomes the low signal-to-noise ratio issue in input signal prevalent in conventional methods. This advancement will benefit the development of BC mixing state observations and radiative effect assessments. Overall, the manuscript is well-organized, and I recommend its publication after minor revisions.*

**Response:** We are grateful for the reviewer's valuable comments. We have carefully revised the manuscript by: (1) adding a detailed analysis of the model's performance across different particle diameter ranges, (2) including a comprehensive comparison of inversion results between training and testing sets in the Supplementary Information, and (3) enhancing the introduction section with a detailed rationale for selecting the LightGBM algorithm. Please find our responses (blue text) to the comments (black text) below.

*1) There appears to be a correlation between the deviation of predicted values from the true values and particle size, as observed in Figure 3c. It would be beneficial to further characterize the relationship between prediction accuracy and particle diameter ($D_p$). This analysis could provide valuable insights into the model's performance across different particle size ranges and potentially identify any systematic biases or limitations in the prediction methodology.*

**Response:** Thank you for pointing this out. We have added a comprehensive prediction error analysis for the $D_\mathrm{p}$ inversion model of internally mixed BC to further characterize the relationship between prediction accuracy and $D_\mathrm{p}$. The added analysis can be found on Lines 310 to 323 in the revised manuscript and attached below:

*"To comprehensively assess the model's performance across different particle size ranges, we further analyzed the prediction error distribution for $D_p$ inversion model of internally mixed BC, as shown in Fig. 5. For particles smaller than 150 nm, the prediction errors average around 4 nm, primarily due to the low signal-to-noise ratio of their scattering signals, which introduces larger uncertainties in the LEO fitting process. The model exhibits optimal performance for particles between 150 nm and 300 nm, with an average prediction error of approximately 1.5 nm. Furthermore, based on the 25% and 75% percentiles of the error distribution, the model's prediction errors exhibit minimal fluctuation within this size range. However, prediction errors gradually increase with particle size, becoming particularly significant for particles larger than 480 nm. This trend can be attributed to occasional irregular signals at larger sizes, such as scattering or incandescence signals with abnormally broad peak widths. These signal irregularities pose challenges to the accurate characterization of particle physical properties,*

*affecting both LEO fitting accuracy and ML model predictions, potentially leading to more pronounced discrepancies between the two methods. The number size distribution of internally mixed BC in the testing set indicates that most particles fall within the 150–300 nm range, where the model demonstrates highest accuracy. Although the prediction errors are relatively larger at both ends of the size distribution (< 150 nm and > 400 nm), the number of particles in these ranges is comparatively small, thus having limited impact on the overall performance of the model."*

[Figure]

*"**Figure 5. The prediction error distribution for $D_p$ inversion model of internally mixed BC, and normalized number size distribution for $D_p$ of internally mixed BC in the testing set. The solid lines in error distribution represent the median value, the upper and lower boundary of the is between the 25 % and 75 % quantiles.**"*

*2) Does the deviation between the predicted values and the true values refer to the test set, or does it also occur in the training set? What could be the underlying reasons for this? Please clarify.*

**Response:** Thank you for your question. The observed deviation between predicted and true values exists in both training and testing sets. As shown in Figure R1, the coefficients of determination $R^2$ for the training and testing sets are 0.99 and 0.98, respectively. These high $R^2$ values indicate excellent model performance, with the close $R^2$ values demonstrating the model's strong predictive capability and good generalization performance.

[Figure]

**Figure R5.** The $D_p$ inversion results of internally mixed BC for both training set (a) and testing set (b).

The SP2 measurement process introduces intrinsic variability primarily through instrument background noise and measurement uncertainties. These factors contribute to inevitable errors and challenges in achieving high-precision measurements, even when employing advanced algorithms like LightGBM. Additionally, BC-containing particles exhibit complex signal characteristics marked by non-linear relationships, varying core-shell structures, and signal-to-noise ratio limitations, particularly for both ends of the particle size distribution, as we discussed in our response to Comment 1. These complexities affect both traditional physical inversion methods and machine learning predictions, leading to discrepancies between the "true" values (which have been renamed as "observed values" in the revised manuscript) obtained from physical inversion and the predicted values from machine learning models.

Furthermore, it's important to note that the LEO fitting method and ML method utilize different parts of the original signals, which can lead to discrepancies in $D_p$ values. We conducted a comprehensive comparison between the LEO fitting method and the machine learning method, elaborating on the differences in signal utilization between the two approaches and their impact on inversion results. A detailed discussion can be found on Lines 395 to 402 of the revised manuscript. Moreover, the Fig. R1 and related content have been added as Fig. S3 to the Supplementary Information.

Lines 393 to 400:

*"Figure 9 illustrates the LEO fitting results for two different BC-containing particles. Despite nearly identical leading-edge data resulting in similar Gaussian distributions and consequently the same $D_p$ values through LEO fitting, the complete scattering signals of these particles exhibit significant differences. The ML model, by incorporating these distinctive signal features, can effectively capture these variations, leading to different $D_p$ predictions. Moreover, the leading edge is traditionally defined as the signal from baseline-subtracted zero up to 5 % of the maximum laser intensity (Taylor et al., 2015). As shown in Fig. 9, this portion of the signal (in the grey-shaded area) is close to the baseline, making it particularly susceptible to noise interference. Compared to LEO fitting method, the ML model utilized a broad range of signals with a high signal-to-noise ratio, demonstrating enhanced noise resistance."*

[Figure]

*"**Figure 9.** Comparison of the scattering signal used in the $D_p$ inversion process for internally mixed BC and corresponding calculation results from both the LEO fitting and the ML methods. The solid line represents the scattering signal obtained by SP2, and the part marked with solid dots is the scattering signal input to the ML model. The gray shaded area shows the leading-edge data used in the LEO fitting process, and the dashed line represents the scattering signal of the original particle reconstructed by LEO fitting."*

*3) Does the inversion of $D_c$ and $D_p$ in BC-containing particles utilize multiple outputs from the same trained model, or from different models? Additionally, does $D_c$ influence the inversion of $D_p$?*

**Response:** Thank you for your comment. This study employs different inversion models for $D_p$ and $D_c$ of BC-containing particles. Although both models are founded on the LightGBM algorithm, they are developed independently due to their distinct feature data and target variables. The $D_c$ inversion model uses only incandescence signals as features, while the $D_p$ inversion model incorporates both incandescence and scattering signals. Consequently, two separate models with different hyperparameters are required to establish unique mappings between their respective input signals and particle characteristics.

For internally mixed BC, $D_c$ influences the inversion of $D_p$. In the traditional physical inversion method, we first obtain the peak height of the reconstructed scattering signal through LEO fitting, and then derive the scattering cross-section. This scattering cross-section is determined by both the BC core and coating material. Consequently, $D_c$ needs to be integrated with Mie theory to accurately estimate $D_p$. A detailed introduction of this method is provided in the "Construction of label dataset" section of the revised manuscript.

In our machine learning method, we similarly consider the influence of $D_c$. The

characteristics of the BC core in internally mixed BC are reflected in the incandescence signal. Therefore, when constructing the $D_p$ inversion model for internally mixed BC, we incorporate both scattering signals and incandescence signals as feature data. This methodology allows us to capture the influence of both the BC core and the coating on the overall particle size, enabling a more accurate prediction of $D_p$ for BC-containing particles.

Lines 208 to 215:

*"As the evaporation of the particle, the scattering signal deviates from a Gaussian distribution, making it inappropriate to directly use the scattering amplitude to calculate $D_p$. **To properly size these particles, the LEO fitting method is employed to reconstruct the Gaussian signal. As described in Sect. 3.3, the zero-crossing point in the TEAPD signal can serve as a position reference for particles in the SP2. Moreover, the position difference between the zero-crossing point and the peak laser intensity remains constant during measurements. The width of the laser intensity distribution and the position of peak laser intensity relative to the zero-crossing point, both determined by Gaussian fitting of numerous unsaturated pure scattering particles, are used to constrain the LEO fitting, leaving the fitting amplitude as the only free parameter. Using leading-edge data from the signal onset to 5% of the maximum laser intensity for LEO fitting, can obtain the reconstructed scattering amplitude and further convert it to particle scattering cross-section.** The $D_p$ of internally mixed BC can be derived by inputting the LEO-fitted scattering cross-section, BC core diameter, and the corresponding refractive indices of the core and coating into the Mie calculation model (Laborde et al., 2012; Liu et al., 2014; Schwarz et al., 2008; Taylor et al., 2015)."*

*4) Could you elaborate on the rationale behind selecting LightGBM over alternative models?*

**Response:** Thank you for your suggestion. We have provided a more comprehensive rationale for selecting LightGBM over alternative models in the introduction of the revised manuscript.

The relevant amendments are detailed on Lines 55 to 66:

[revised manuscript text omitted]